# A Holistic View of Label Noise Transition Matrix in Deep Learning and Beyond

**Yong Lin**[1*]  **Renjie Pi**[1*]  **Weizhong Zhang**[2]  **Xiaobo Xia**[3]  **Jiahui Gao**[4]  **Xiao Zhou**[1]
**Tongliang Liu**[3]    **Bo Han**[5]

[1]The Hong Kong University of Science and Technology   [2]Fudan University
[3]Sydney AI Centre, The University of Sydney   [4]The University of Hong Kong
[5]Hong Kong Baptist University

## Abstract

In this paper, we explore learning statistically consistent classifiers under label noise by estimating the noise transition matrix ($T$). We first provide a holistic view of existing $T$-estimation methods including those with or without anchor point assumptions. We unified them into the *Minimum Geometric Envelope Operator* (MGEO) framework, which tries to find the smallest $T$ (in terms of a certain metric) that elicits a convex hull to enclose the posteriors of all the training data. Although MGEO methods show appealing theoretical properties and empirical results, we find them prone to failing when the noisy posterior estimation is imperfect, which is inevitable in practice. Specifically, we show that MGEO methods are in-consistent even with infinite samples if the noisy posterior is not estimated accurately. In view of this, we make the first effort to address this issue by proposing a novel $T$-estimation framework via the lens of bilevel optimization, and term it *RObust Bilevel OpTimzation* (ROBOT). ROBOT paves a new road beyond MGEO framework, which enjoys strong theoretical properties: identifibility, consistency and finite-sample generalization guarantees. Notably, ROBOT neither requires the perfect posterior estimation nor assumes the existence of anchor points. We further theoretically demonstrate that ROBOT is more robust in the case where MGEO methods fail. Experimentally, our framework also shows superior performance across multiple benchmarks. Our code is released at https://github.com/pipilurj/ROBOT [†].

## 1 Introduction

Deep learning has achieved remarkable success in recent years, owing to the availability of abundant computational resources and large scale datasets to train the models with millions of parameters. Unfortunately, the quality of datasets can not be guaranteed in practice. There are often a large amount of mislabelled data in real world datasets, especially those obtained from the internet through crowd sourcing (Li et al., 2021; Xia et al., 2019; 2020; Xu et al., 2019; Wang et al., 2019; 2021; Liu et al., 2020; Collier et al., 2021; Bahri et al., 2020; Li et al., 2022a; Yong et al., 2022; Lin et al., 2022; Zhou et al., 2022b). This gives rise to the interest in performing learning under label noise.

Our task is to learn a function $f_\theta$ with parameter $\theta$ to predict the clean label $Y \in \mathcal{Y} = \{1, ..., K\}$ based on the input $\boldsymbol{X} \in \mathcal{X} = \mathbb{R}^d$. However, we only observe a noisy label $\tilde{Y}$ which is generated from $Y$ by an (oracle) noisy transition matrix $\boldsymbol{T}^*(\boldsymbol{x})$ whose element $T_{ij}^*(\boldsymbol{x}) = P(\tilde{Y} = j | Y = i, \boldsymbol{X} = \boldsymbol{x})$. We consider class-dependent label noise problems which assume that $\boldsymbol{T}^*$ is independent of $\boldsymbol{x}$, i.e., $\boldsymbol{T}^*(\boldsymbol{x}) = \boldsymbol{T}^*$. Given $\boldsymbol{T}^*$, we can obtain the clean posterior from the noise posterior by $P(Y|\boldsymbol{X} = \boldsymbol{x}) = (\boldsymbol{T}^*)^{-1} P(\tilde{Y}|\boldsymbol{X} = \boldsymbol{x})$. With the oracle $\boldsymbol{T}^*$, we can learn a *statistically consistent* model on the noisy dataset which collides with the optimal model for the clean dataset. Specifically, denote the $\theta^*$ as the minimizer of the clean loss $\mathbb{E}_{\boldsymbol{X},Y}[\ell(f_\theta(\boldsymbol{X}), Y)]$ where $\ell$ is the cross entropy loss. Then minimizing the $\mathbb{E}_{\boldsymbol{X},\tilde{Y}}[\ell(\boldsymbol{T}^* f_\theta(\boldsymbol{X}), \tilde{Y})]$ also leads to $\theta^*$. Therefore, the effectiveness of such methods heavily depends on the quality of estimated $\boldsymbol{T}$.

---

[*]Equal Contribution, correspondence to Renjie Pi (rpi@connect.ust.hk)
[†]Code released at https://github.com/pipilurj/ROBOT

To estimate $\boldsymbol{T}^*$, earlier methods assume that there exists anchor points which belong to a certain class with probability one. The noisy posterior probabilities of the anchor points are then used to construct $\boldsymbol{T}^*$ (Patrini et al., 2017; Liu & Tao, 2015). Specifically, they fit a model on the noisy dataset and select the most confident samples as the anchor points. However, as argued in (Li et al., 2021), the violation of the anchor point assumption can lead to inaccurate estimation of $\boldsymbol{T}$. To overcome this limitation, Li et al. (2021); Zhang et al. (2021) then try to develop anchor-free methods that are able to estimate $\boldsymbol{T}$ without the anchor point assumption (See Appendix C for more discussion on related work).

Since $P(\tilde{Y}|\boldsymbol{X} = \boldsymbol{x}) = \boldsymbol{T}^* P(Y|\boldsymbol{X} = \boldsymbol{x})$ and $\sum_{i=1}^{C} P(Y = i|\boldsymbol{X} = \boldsymbol{x}) = 1$, we know that for any $\boldsymbol{x}$, $P(\tilde{Y}|\boldsymbol{X} = \boldsymbol{x})$ is enclosed in the convex hull conv$\{\boldsymbol{T}^*\}$ formed by the columns of $\boldsymbol{T}^*$. Therefore, both anchor-based and anchor-free methods try to find a $\boldsymbol{T}$ whose conv$\{\boldsymbol{T}\}$ encloses the noisy posteriors of all data points, under the assumption that all the noisy posteriors are perfectly estimated. To identify $\boldsymbol{T}^*$ from all the $\boldsymbol{T}$s satisfying the above condition, they choose the smallest one in terms of certain positive metrics, e.g., anchor-based methods adopts equation 2 and minimum volume (anchor-free) adopts equation 3. Therefore, we unify them into the framework of Minimum Geometric Envelope Operator (MGEO), the formal definition of which is in Section 2. Though MGEO-based methods achieve remarkable success, we show that they are sensitive to noisy posteriors estimation errors. Notably, neural networks can easily result in inaccurate posterior estimations due to their over-confidence (Guo et al., 2017). If some posterior estimation errors skew the smallest convex hull that encloses all the data points, MGEO can result in an unreliable $\boldsymbol{T}$-estimation returned (as illustrated in Fig. 1 (a)). We theoretically show that even if the noisy posterior is accurate except for a single data point, MGEO-based methods can result in a constant level error in $\boldsymbol{T}$-estimation. We further provide supportive experimental results for our theoretical findings in Section 2.

In view of this, we aim to go beyond MGEO by proposing a novel framework for stable end-to-end $\boldsymbol{T}$-estimation. Let $\hat{\theta}(\boldsymbol{T})$ be the solution of $\theta$ to minimize $\mathbb{E}_{\boldsymbol{X},\tilde{Y}}[\ell(\boldsymbol{T}f_\theta(\boldsymbol{X}), \tilde{Y})]$ when $\boldsymbol{T}$ is fixed. Here $\hat{\theta}(\boldsymbol{T})$ explicitly shows the returned $\hat{\theta}$ depends on $\boldsymbol{T}$. If the clean dataset is available, we can find $\boldsymbol{T}^*$ by checking whether $\boldsymbol{T}$ induces a $\hat{\theta}(\boldsymbol{T})$ that is optimal for $\mathbb{E}_{\boldsymbol{X},Y}[\ell(f_\theta(\boldsymbol{X}), Y)]$ (this is ensured by the consistency of forward correction method under suitable conditions (Patrini et al., 2017), which is discussed in Appendix A.3). Then the challenge arises, as we do not have the clean dataset in practice. Fortunately, we have the well established robust losses (denoted as $\ell_{\text{rob}}$), e.g., Mean Absolute Error (MAE) (Ghosh et al., 2017) and Reversed Cross Entropy (RCE) (Wang et al., 2019) whose minimizer on $\mathbb{E}_{\boldsymbol{X},\tilde{Y}}[\ell_{\text{rob}}(f_\theta(\boldsymbol{X}), \tilde{Y})]$ collides with that on $\mathbb{E}_{\boldsymbol{X},Y}[\ell_{\text{rob}}(f_\theta(\boldsymbol{X}), Y)]$. Therefore, we search for $\boldsymbol{T}^*$ by checking whether $\boldsymbol{T}$ minimizes $\mathbb{E}_{\boldsymbol{X},\tilde{Y}}[\ell_{\text{rob}}(f_{\hat{\theta}(\boldsymbol{T})}(\boldsymbol{X}), \tilde{Y})]$, which only depends on the noisy data. This procedure can be naturally formulated as a bilevel problem as follows: in the inner loop, $\boldsymbol{T}$ is fixed and we obtain $\hat{\theta}(\boldsymbol{T})$ by training $\theta$ to minimize $\mathbb{E}_{\boldsymbol{X},\tilde{Y}}[\ell(\boldsymbol{T}f_\theta(\boldsymbol{X}), \tilde{Y})]$. In the outer loop, we train $\boldsymbol{T}$ to minimize $\mathbb{E}_{\boldsymbol{X},\tilde{Y}}[\ell(\boldsymbol{T}f_{\hat{\theta}(\boldsymbol{T})}(\boldsymbol{X}), \tilde{Y})]$. We named our framework as RObust Bilevel OpTmization (ROBOT).

Notably, different from MGEO, ROBOT is based on sample mean estimator which is intrinsically consistent by the law of large numbers. In Section 3.2, we show the theoretical properties of ROBOT: identifiablilty, finite sample generalization and consistency. Further, ROBOT achieves $O(1/n)$ robustness to the noisy posterior estimation error in the case where MGEO methods lead to a constant level $\boldsymbol{T}$-estimation error. In Section 4, we conduct extensive experiments on both synthetic and real word datasets. Our methods beat the MGEO methods significantly both in terms of prediction accuracy and $\boldsymbol{T}$-estimation accuracy.

**Contribution**.

- We provide the first framework MGEO to unify the existing $\boldsymbol{T}$-estimation methods including both anchor-based and anchor-free methods. Through both theoretically analysis and empirical evidence, we formally identify the instability of MGEO-based methods when the noisy posteriors are not estimated perfectly, which is inevitable in practice due to the over-confidence of large neural networks.

- To break through the limitation of MGEO-based methods, we propose a novel framework ROBOT to estimate $\boldsymbol{T}$ that only relies on sample mean estimators, which is consistent by the law of large numbers. ROBOT enjoys strong theoretical guarantees including identifi-bility, consistency and finite sample generalization without assuming perfect noisy posteri-

ors or the existence of anchor points. Further, when the posterior estimation is imperfect, the error of ROBOT is bounded $O(1/n)$ while that of MGEO is in constant level.

- Extensive experiments over various popular benchmarks show that ROBOT improves over MGEO-based methods by a large margin in terms of both test accuracy and $\boldsymbol{T}$ estimation error. For instance, ROBOT increases $\sim 10\%$ accuracy and decreases $\sim 40\%$ $\boldsymbol{T}$-estimation error over MGEO-based methods on CIFAR100 with uniform noisy.

## 2    MINIMUM GEOMETRIC ENVELOPE OPERATOR (MGEO)

**Preliminaries.** Throughout this paper, we use upper cased letters, i.e., $\boldsymbol{X}$, to denote random vectors, use $\boldsymbol{x}$ to denote deterministic scalars and vectors. For any vector $\boldsymbol{v}$, we use $\boldsymbol{v}[i]$ or $\boldsymbol{v}_i$ to denote the $i$th element of $\boldsymbol{v}$. The L1 norm of $\boldsymbol{v}$ is $|\boldsymbol{v}|_1 = \sum_{i=1}^{d} |v_i|$. Let $\boldsymbol{T}^*$ denote the oracle noise transition matrix as introduced in Section 1. Let $T_{i\cdot}$ and $T_{\cdot j}$ be the $i$th row and $j$th of $\boldsymbol{T}$ column, respectively. Denote the feasible region of $\boldsymbol{T}$ as $\mathcal{T} := \{\boldsymbol{T}|T_{ij} > 0, |T_{\cdot j}|_1 = 1, \forall i, j \in [K]\}$. Let $\boldsymbol{e}^i \in \mathbb{R}^K$ denote the unit vector where $\boldsymbol{e}^i[i] = 1$. $\mathcal{D}(n)$ and $\tilde{\mathcal{D}}(n)$ denote the clean and noisy dataset with $n$ samples, respectively; let $\mathcal{D}$ and $\tilde{\mathcal{D}}$ denote $\mathcal{D}(\infty)$ and $\tilde{\mathcal{D}}(\infty)$ for short. The risk on dataset $\mathcal{D}(n)$ is $\mathcal{L}(\theta, \mathcal{D}(n)) := \frac{1}{n} \sum_{(\boldsymbol{x},y) \in \mathcal{D}(n)} [\ell(f_\theta(\boldsymbol{x}), y)]$ where $\ell$ is the loss function. By default, we use cross entropy as $\ell$. Denote the noisy posterior $P(\tilde{Y}|\boldsymbol{X} = \boldsymbol{x})$ as $\tilde{g}(\boldsymbol{x})$ for short. Let $g(\boldsymbol{x})$ denote the fitted posterior that we obtain. Let $\mathcal{G}(n)$ denote the set of the fitted posteriors, i.e., $\mathcal{G}(n) := \{g(\boldsymbol{x}_i)\}_{i=1}^n$.

### 2.1    MGEO AND ITS LIMITATION

In this section, we first unify the existing works of $\boldsymbol{T}$-estimation by MGEO. Denote the convex hull induced by the columns $\boldsymbol{T}$ as $\text{conv}(\boldsymbol{T}) = \{\boldsymbol{t}|\boldsymbol{t} = \sum_{i=1}^K \alpha_i \boldsymbol{t}_i, \sum_{i=1}^K \alpha_i = 1, \alpha_i > 0\}$, where $\boldsymbol{T} = [\boldsymbol{t}_1...\boldsymbol{t}_K]$. Notably, $\text{conv}(\boldsymbol{T})$ is the feasible region of $\tilde{g}(\boldsymbol{x})$ generated by $\boldsymbol{T}$ because $\tilde{g}(\boldsymbol{x}) = P(\tilde{Y}|\boldsymbol{X} = \boldsymbol{x}) = \boldsymbol{T}P(Y|\boldsymbol{X} = \boldsymbol{x})$ and $|P(Y|\boldsymbol{X} = \boldsymbol{x})|_1 = 1$. Existing methods assume that we can fit the posteriors perfectly, i.e., $g(\boldsymbol{x}) = \tilde{g}(\boldsymbol{x})$ for all $\boldsymbol{x}$. They then try to find a $\boldsymbol{T}$ whose $\text{conv}(\boldsymbol{T})$ encloses $\mathcal{G}(n)$ (Patrini et al., 2017; Zhang et al., 2021; Xia et al., 2020; Li et al., 2021; Zhang et al., 2021; Liu & Tao, 2015). Since there are infinitely many $\boldsymbol{T}$ satisfying this conditions, they chose the smallest one in terms of certain positive metrics. We name them as the Minimum Geometric Envelope Operator (MGEO) and present the unified definition of them as follows:

**Definition 1** (Minimum Geometric Envelope Operator)**.** *An operator $Q : \mathbb{R}^{K \times n} \to \mathcal{T}$ on a set $\mathcal{G}(n) = \{g(\boldsymbol{x}_i)\}_{i=1}^n$ is said to be Minimum Geometric Envelope Operator (MGEO) if it solves*

$$\boldsymbol{T}^{\text{MGEO}} = \underset{\boldsymbol{T} \in \mathcal{T}}{\arg\min} \, \mathcal{M}(\boldsymbol{T}), \quad \text{s.t. } \mathcal{G}(n) \subset \text{conv}(\boldsymbol{T}), \tag{1}$$

*where $\boldsymbol{T} = [\boldsymbol{t}_1...\boldsymbol{t}_K]$ and $\text{conv}(\boldsymbol{T}) = \{\boldsymbol{t}|\boldsymbol{t} = \sum_{i=1}^K \alpha_i \boldsymbol{t}_i, \sum_{i=1}^K \alpha_i = 1, \alpha_i > 0\}$. We denote it as $\boldsymbol{T}^{\text{MGEO}} = Q(\mathcal{G}(n))$ for short.*

Now we proceed to show how MGEO takes anchor-based and anchor-free methods as special cases (a brief description of these methods is included in Appendix A.1).

**Example 1 (anchor-based).** For each class $j \in [K]$, anchor-based methods assumes that there exists an anchor point $\boldsymbol{x}^j$ with $P(Y|\boldsymbol{X} = \boldsymbol{x}^j) = \boldsymbol{e}^j$. So $T_{\cdot j} = \tilde{g}(\boldsymbol{x}^j)$ and $\tilde{g}(\boldsymbol{x}^j)[j] \geq \tilde{g}(\boldsymbol{x})[j]$ for all $\boldsymbol{x}$. Anchor methods first fit $g(\cdot)$ on the noisy data and then find the most confident sample $\boldsymbol{x}^j$ in $\mathcal{G}(n)$ for class $j$. Then they set $T_{\cdot j} = g(\boldsymbol{x}^j)$. This is equivalent to solving equation 1 with the following metric:

$$\mathcal{M}(\boldsymbol{T}) = \sum_{j \in [K]} \left( T_{jj} + \min_{i \in [n]} \|T_{\cdot j} - g(\boldsymbol{x}_i)\|_2 \right). \tag{2}$$

The intuition is as follows: because equation 1 requires $\mathcal{G}(n)$ to be contained in $\text{conv}(\boldsymbol{T})$, we must have $T_{jj} \geq g(\boldsymbol{x}^j)[j]$ or otherwise $g(\boldsymbol{x}^j)$ will be outside of $\text{conv}(\boldsymbol{T})$. At the same time, we have $\min_{i \in [n]} \|T_{\cdot j} - g(\boldsymbol{x}_i)\|_2 \geq 0$ for all $\boldsymbol{T}$ and $j$. So choosing $T_{\cdot j} = g(\boldsymbol{x}^j)$ can minimize equation 2 because $T_{jj} = g(\boldsymbol{x}^j)[j]$ and $\|T_{\cdot j} - g(\boldsymbol{x}^j)\|_2 = 0$.

**Example 2 (Minimum Volume, anchor-free).** The Minimum Volume method proposed by Li et al. (2021) uses the volume of $\boldsymbol{T}$ as the metric:

$$\mathcal{M}(\boldsymbol{T}) = \text{Vol}(\boldsymbol{T}). \tag{3}$$

The MGEO-based methods can work well if the posterior is perfectly estimated, i.e., $g(\boldsymbol{x}) = \tilde{g}(\boldsymbol{x})$ for all $\boldsymbol{x}$. In this case, MGEO methods can identify $\boldsymbol{T}^*$ under suitable conditions. However, it is common that $g(\boldsymbol{x}) \neq \tilde{g}(\boldsymbol{x})$ for some $\boldsymbol{x}$ in practice because of DNN's over-confidence. An error in the posterior estimation can easily skew the smallest convex hull which encloses all data points as illustrated in Figure 1. Then MGEO methods lead to inaccurate $\boldsymbol{T}$-estimation in this case. To understand the sensitivity of the MEGO methods to the posterior estimation error, we consider a simple case where $g(\boldsymbol{x})$ agrees with $\tilde{g}(\boldsymbol{x})$ almost everywhere except from a single point $\boldsymbol{x}' \in \mathcal{D}(n)$:

$$g_{\epsilon, \boldsymbol{x}'}(\boldsymbol{x}) = \begin{cases} \tilde{g}(\boldsymbol{x}) + \epsilon, & \text{if} \quad \boldsymbol{x} = \boldsymbol{x}', \\ \tilde{g}(\boldsymbol{x}), & \text{otherwise.} \end{cases} \tag{4}$$

where $\epsilon$ is the error of the estimated posterior at $\boldsymbol{x}'$ and it needs to satisfy that $g(\boldsymbol{x}') + \epsilon$ has all non-negative elements which sum up to 1, i.e., $\epsilon \in \Xi := \{\epsilon | |\epsilon + \tilde{g}(\boldsymbol{x}')|_1 = 1, (\epsilon + \tilde{g}(\boldsymbol{x}'))[i] \geq 0, \forall i \in [K]\}$. Let $\mathcal{G}_{\epsilon, \boldsymbol{x}'}(n) := \{g_{\epsilon, \boldsymbol{x}'}(\boldsymbol{x}_i)\}_{i=1}^n$ be set of the fitted posteriors by $g_{\epsilon, \boldsymbol{x}'}(\cdot)$. Similarly, we denote $\tilde{\mathcal{G}}(n) := \{\tilde{g}(\boldsymbol{x}_i)\}_{i=1}^n$. Further, let $\boldsymbol{T}^{\text{MGEO}}$ and $\boldsymbol{T}_{\epsilon, \boldsymbol{x}'}^{\text{MGEO}}$ be the solution of MGEO on $\tilde{\mathcal{G}}(n)$ and $\mathcal{G}_{\epsilon, \boldsymbol{x}'}(n)$, i.e., $\boldsymbol{T}^{\text{MGEO}} := Q(\tilde{\mathcal{G}}(n))$ and $\boldsymbol{T}_{\epsilon, \boldsymbol{x}'}^{\text{MGEO}} := Q(\mathcal{G}_{\epsilon, \boldsymbol{x}'}(n))$. We have $\boldsymbol{T}^{\text{MGEO}} = \boldsymbol{T}^*$ under suitable conditions (Patrini et al., 2017; Li et al., 2021). We are then interested in the $\boldsymbol{T}$-estimation error due to $\epsilon$ in terms of the Frobenius norm, i.e., $\|\boldsymbol{T}_{\epsilon, \boldsymbol{x}'}^{\text{MGEO}} - \boldsymbol{T}^*\|_{\text{F}}$. Following are the results:

**Proposition 1.** *Under assumptions specified in Appendix A.4, suppose that we obtain an imperfect estimation of the noise posterior $g_{\epsilon, \boldsymbol{x}'}(\cdot)$ as described in equation 4. Then MGEO-based methods lead to a $\boldsymbol{T}$-estimation error whose minimax lower bound is:* $\sup_{\epsilon \in \Xi} \|\boldsymbol{T}_{\epsilon, \boldsymbol{x}'}^{\text{MGEO}} - \boldsymbol{T}^*\|_{\text{F}} \geq \Omega(1)$.

See Appendix A.4 for the proof. Proposition 1 shows that MEGO methods can lead to a constant level of $\boldsymbol{T}$-estimation error under the fitted posterior $g_{\epsilon, \boldsymbol{x}'}(\cdot)$. This is in analogy to Figure 1 (left) that a single outlier caused by $\epsilon$ can skew the smallest convex hull that encloses all samples. Note that Proposition 1 shows that the error caused by the inaccurate posterior estimation does not shrink to zero as the sample size increases. Thus MGEO methods can be in-consistent in this case. In the following corollary of Proposition 1, we formally state the in-consistency of MEGO:

**Corollary 1** (In-consistency of MGEO)**.** *Under assumptions specified in Appendix A.4, suppose that we obtain an imperfect estimation of the noise posterior $g_{\epsilon, \boldsymbol{x}'}(\cdot)$ as described in equation 4. Then MGEO-based methods can be inconsistent, i.e., there exists an $\epsilon$ such that $\boldsymbol{T}_{\epsilon, \boldsymbol{x}'}^{\text{MGEO}} \not\to \boldsymbol{T}^*$ as $n \to \infty$.*

*Proof.* In the proof of Proposition 1, we already show that there exists an $\epsilon$ such that $\|\boldsymbol{T}_{\epsilon, \boldsymbol{x}'}^{\text{MGEO}} - \boldsymbol{T}^*\|_{\text{F}} \geq \Omega(1)$. Since this result holds for any simple size $n$. If $\lim_{n \to \infty} \boldsymbol{T}_{\epsilon, \boldsymbol{x}'}^{\text{MGEO}}$ does not exists , the claim holds immediately. Otherwise if $\lim_{n \to \infty} \boldsymbol{T}_{\epsilon, \boldsymbol{x}'}^{\text{MGEO}}$ exists, we have $\| \lim_{n \to \infty} \boldsymbol{T}_{\epsilon, \boldsymbol{x}'}^{\text{MGEO}} - \boldsymbol{T}^*\| \geq \Omega(1)$, which leads to $\lim_{n \to \infty} \boldsymbol{T}_{\epsilon, \boldsymbol{x}'}^{\text{MGEO}} \neq \boldsymbol{T}^*$. So we conclude that $\boldsymbol{T}_{\epsilon, \boldsymbol{x}'}^{\text{MGEO}} \not\to \boldsymbol{T}^*$ as $n \to \infty$. $\square$

### 2.2 Empirical Findings

Unfortunately, due to the tendency of Deep Neural Network to make over-confident prediction, there are inevitably outliers due to erroneous posterior estimation. Because MGEO methods attempts to find the smallest $\boldsymbol{T}$ that convers all the samples, those outliers can severely skew the $\boldsymbol{T}$ and degrades the estimation accuracy. We observe this phenomenon widely exists in existing methods. In Figure 1 (b), we illustrate the $\boldsymbol{T}$ estimation results of the Minimum Volume method. The red triangle is the oracle $\boldsymbol{T}^*$. We can see that there are lots of samples whose estimated posteriors are out of the red triangle. The purple triangle are the fitted $\boldsymbol{T}$ by the Minimum Volume method, which is highly skewed by the outliers. Refer to Appendix B.1 for experimental details.

## 3 RObust BIlevel OpTimization (ROBOT)

In the last section, we discussed the instability of MGEO methods when the noisy posterior estimation is imperfect. Proposition 1 shows that even if the estimated posterior differs from $\tilde{g}(\cdot)$ at a single point, MGEO can lead to a constant level of $\boldsymbol{T}$-estimation error. This is because the $\boldsymbol{T}$-estimation of MGEO depends on finding a convex hull to enclose all samples and such convex hull is determined by the the outermost samples. Therefore, if there is an outlier due to inaccurate posterior estimation, the resulting convex hull can be easily skewed.

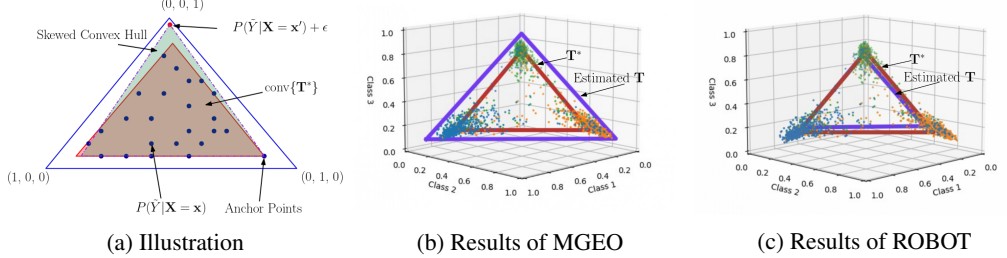

| (a) Illustration | (b) Results of MGEO | (c) Results of ROBOT |

Figure 1: (a) Illustration of how the posterior estimation error can lead to $\boldsymbol{T}$-estimation error in a 3-class classification task. The posteriors of the blue points are accurately estimated and they are all in $\mathrm{conv}(\boldsymbol{T}^*)$. There is an error $\epsilon$ on the noisy posterior of $\boldsymbol{x}'$, which is denoted in red. Because $P(\tilde{Y}|\boldsymbol{X} = \boldsymbol{x}') + \epsilon$ lies outside of $\mathrm{conv}(\boldsymbol{T}^*)$, MGEO needs to find a larger $\boldsymbol{T}$ to enclose it. (b) Visualization of MGEO (Minimum Volume) method in a 3-class MNIST classification task. $\mathrm{conv}(\boldsymbol{T}^*)$ is denoted by the red triangle. There are many outliers whose estimated posteriors are out of $\mathrm{conv}(\boldsymbol{T}^*)$ due to overfitting. MGEO methods result in leads to a inaccurate $\boldsymbol{T}$ because they try to enclose all the samples, including the outliers. (c) ROBOT obtains a more accurate $\boldsymbol{T}$ than MGEO.

Then a natural question to ask is, *can we go beyond Geometric Envelope Operator to obtain a robust and consistent $\boldsymbol{T}-$estimation?* As a thought experiment, let's use a simple case for example. Suppose $\tilde{g}(\boldsymbol{x}_i)[j] = 0.5$ for all $i$. We aim to estimate the $\boldsymbol{T}$ by the anchor point method. If $g(\cdot) = \tilde{g}(\cdot)$, we have $T_{jj} = \max_{i\in[n]} \tilde{g}(\boldsymbol{x}_i)[j] = 0.5$. Now suppose that we obtain a inaccurate posterior with some small Gaussian error, i.e., $g(\cdot)[j] = \tilde{g}(\cdot)[j] + \xi$ where $\xi \sim \mathcal{N}(0, \sigma^2)$ for some small $\sigma$. We then have $\mathbb{E}[T_{jj}] = \mathbb{E}[\max_{i\in[n]} \tilde{g}(\boldsymbol{x}_i)[j] + \xi_i] \to 0.5 + \sigma\sqrt{2\log(n)}$ for large $n$ (Ho & Hsing, 1996). So this method is inconsistent even with infinite samples. On the other hand, it is well known that the sample mean estimator is consistent by $1/n \sum_{i=1}^{n} g(\boldsymbol{x}_i)[j] \to 0.5$ as $n \to \infty$. This toy example shows that the sample mean estimators are consistent while MGEO methods are not because they depends on the maximum of the samples. In light of this, we provide the framework ROBOT as the first attempt to estimate $\boldsymbol{T}$ based on the sample mean estimators.

### 3.1 METHODS

Denote the optimal parameter that minimizes the cross entropy loss on the clean dataset as $\theta^* := \arg\min_{\theta} \mathcal{L}(f_\theta, \mathcal{D})$. A popular method is to minimize following forward correction loss on the noisy dataset (Patrini et al., 2017):

$$\hat{\boldsymbol{T}}, \hat{\theta} = \arg\min_{\boldsymbol{T},\theta} \mathcal{L}(\boldsymbol{T}f_\theta, \tilde{\mathcal{D}}) \tag{5}$$

Patrini et al. (2017) shows that the forward correction is consistent, i.e., $\hat{\theta} = \theta^*$ if $\boldsymbol{T} = \boldsymbol{T}^*$, under suitable conditions. These conditions are mainly on the sufficiently large function space of $f_\theta$ and proper composite loss functions. We discuss these conditions in Appendix A.3 for completeness. Though equation 5 is consistent, solving equation 5 can not uniquely identify $\boldsymbol{T}^*$ and $\theta^*$. This is because if $\boldsymbol{T}^* = \boldsymbol{T}_1\boldsymbol{T}_2$, and $f_{\theta_1}(\cdot) = \boldsymbol{T}_2 f_{\theta^*}(\cdot)$, $\boldsymbol{T}_1$ and $\theta_1$ can also achieve the optimal loss. In order to make $\boldsymbol{T}$ identifiable, MGEO tries to find the minimum $\boldsymbol{T}$ whose $\mathrm{conv}(\boldsymbol{T})$ contains all data points. Now we try to go beyond MGEO and aim to identify $\boldsymbol{T}^*$ by seeking for some sample mean estimators.

Since the solution of $\hat{\theta}$ in equation 5 depends on $\boldsymbol{T}$ if $\boldsymbol{T}$ is fixed, we then use $\hat{\theta}(\boldsymbol{T})$ to explicitly denote such dependency, i.e., $\hat{\theta}(\boldsymbol{T}) = \arg\min_{\theta} \mathcal{L}(\boldsymbol{T}f_\theta, \tilde{\mathcal{D}})$. By the consistency of the forward correction, we already know $\hat{\theta}(\boldsymbol{T}^*) = \theta^*$ under suitable conditions described in Appendix A.3. Therefore, we can seek for a good $\boldsymbol{T}$ by evaluating $\hat{\theta}(\boldsymbol{T})$. If we have the clean dataset $\mathcal{D}$, this would be straightforward by checking whether $\hat{\theta}(\boldsymbol{T})$ minimizes the clean loss $\mathcal{L}(f_{\hat{\theta}(\boldsymbol{T})}, \mathcal{D})$. Specifically, combining $\theta^* = \hat{\theta}(\boldsymbol{T}^*)$ and $\theta^* = \arg\min_{\theta} \mathcal{L}(f_\theta, \mathcal{D})$, we can uniquely identify $\boldsymbol{T}^*$ by $\boldsymbol{T}^* = \arg\min_{\boldsymbol{T}} \mathcal{L}(f_{\hat{\theta}(\boldsymbol{T})}, \mathcal{D})$. Notably, $\mathcal{L}(f_{\hat{\theta}(\boldsymbol{T})}, \mathcal{D})$ depends on the sample mean of the losses, which is consistent according to the law of large numbers. Now the new challenge arises: we do not have the clean dataset $\mathcal{D}$ in practice. Can we find a sample mean estimator only based on the noisy dataset $\tilde{\mathcal{D}}$ which is minimized by $\theta^*$ (which is also $\hat{\theta}(\boldsymbol{T}^*)$)?

Readers familiar with the noise robust losses may already guess our next proposal. Existing works have proposed several losses robust to label noise, e.g., Mean Absolute Error (MAE) and Reversed Cross-entropy (RCE), whose minimizer is the same on the noisy dataset with that on the clean dataset under suitable conditions (Wang et al., 2019; Ghosh et al., 2017). These conditions are described in Appendix A.2. Let $\ell_{\text{rob}}$ be the noise robust losses and $\mathcal{L}_{\text{rob}}$ be the corresponding risk on datasets. Then we have the following:

$$\arg\min_{\theta} \mathcal{L}_{\text{rob}}(f_{\theta}, \tilde{\mathcal{D}}) = \arg\min_{\theta} \mathcal{L}_{\text{rob}}(f_{\theta}, \mathcal{D}). \tag{6}$$

Therefore, we can use $\mathcal{L}_{\text{rob}}(f_{\theta}, \tilde{\mathcal{D}})$ to measure the optimality of $\hat{\theta}(\boldsymbol{T})$. This property enables us to seek for $\boldsymbol{T} = \boldsymbol{T}^*$ by checking whether $\boldsymbol{T}$ minimizes $\mathcal{L}_{\text{rob}}(f_{\hat{\theta}(\boldsymbol{T})}, \tilde{\mathcal{D}})$. The above-mentioned procedure can be naturally formulated into a bi-level problem: We split the noisy dataset into a training set $\tilde{\mathcal{D}}_{tr}$ and a validation set $\tilde{\mathcal{D}}_v$. In the inner loop, $\boldsymbol{T}$ is fixed and we obtain $\hat{\theta}(\boldsymbol{T})$ by minimizing the forward correction loss on the training set over $\theta$, i.e., $\hat{\theta}(\boldsymbol{T}) := \arg\min_{\theta} \mathcal{L}(\boldsymbol{T}f_{\theta}, \tilde{\mathcal{D}}_{tr})$. In the outer loop, we optimize the robust loss of $f_{\hat{\theta}(\boldsymbol{T})}$ on the validation dataset by minimizing over $\boldsymbol{T}$, i.e., $\min_{\boldsymbol{T}} \mathcal{L}_{\text{rob}}(f_{\hat{\theta}(\boldsymbol{T})}, \tilde{\mathcal{D}}_v)$. We summarize the bilevel procedure as follows:

$$\min_{\boldsymbol{T}} \mathcal{L}_{\text{rob}}(f_{\hat{\theta}(\boldsymbol{T})}, \tilde{\mathcal{D}}_v) \tag{7}$$
$$\text{s.t. } \hat{\theta}(\boldsymbol{T}) = \arg\min_{\theta} \mathcal{L}(\boldsymbol{T}f_{\theta}, \tilde{\mathcal{D}}_{tr}).$$

We name it as RObust Bilvel OpTmization (ROBOT). Remarkably, both $\mathcal{L}$ and $\mathcal{L}_{\text{rob}}$ are consistent estimators by the law of large numbers without requiring perfect noisy posteriors (Jeffreys, 1998). We also include the convergence of ROBOT in Appendix A.9 for completeness, which is an application of standard bilevel optimization.

**Remark 1.** *A curious reader may wonder if it is possible to use a two-step procedure instead of the bilevel framework: first learn $\hat{\theta}$ by minimizing equation 6 and then plug the $\hat{\theta}$ into equation 5 to obtain $\hat{\boldsymbol{T}}$. From the statistical property view (ignoring the optimization difficulty), we may have $\hat{\theta} = \theta^*$ with infinite samples and then it can further leads to $\hat{\boldsymbol{T}} = \boldsymbol{T}^*$. However, existing works show that robust losses are very hard to optimize (Zhang & Sabuncu, 2018; Wang et al., 2019), indicating that directly optimizing equation 6 on $\theta$ can hardly lead to the optimal $\theta^*$ in practice. Our ROBOT in equation 7 transforms the optimization from the space of neural network parameters to the space of $\boldsymbol{T}$ when minimizing $\mathcal{L}_{\text{rob}}(f_{\hat{\theta}(\boldsymbol{T})}, \tilde{\mathcal{D}})$ over $\boldsymbol{T}$. The experimental results in Appendix B.4 show that training ROBOT can significantly decrease the training robust loss while directly optimizing the robust loss fails to do so. This indicates that the reparametrization of ROBOT may have better convergence property on minimizing the robust loss. We will investigate the in-depth mechanism of this interesting phenomenon in the future.*

### 3.2 THEORETICAL ANALYSIS OF ROBOT

In this section, we analyze the theoretical properties of ROBOT in equation 7. First we show the identifiability of $\boldsymbol{T}^*$ with infinite noisy samples. Then we provide the finite sample generalization bound together with the consistency. We first start with some mild assumptions:

**Assumption 1.** *The optimal $\theta^*$ for the cross entropy on the clean dataset also minimizes the robust losses on the clean dataset $\mathcal{D}$, i.e., $\mathcal{L}_{\text{rob}}(\theta^*, \mathcal{D}) < \mathcal{L}_{\text{rob}}(\theta, \mathcal{D})$ for all $\theta \neq \theta^*$.*

This assumption is natural because $f_{\theta^*}(\boldsymbol{x}) = P(Y|\boldsymbol{X} = \boldsymbol{x})$ is supposed to be optimal for both cross entropy and robust losses on the clean dataset. Further, Ghosh et al. (2017) shows that these losses are class-calibrated, and minimizing them leads to the decrease in 0–1 error. We assume $\theta^*$ is unique, or otherwise we can also concern ourselves with the one with the minimum norm and the identifiability results are the same. Notably, robust losses are more difficult to optimize and they often lead to under-fitting when applied on deep neural networks (Wang et al., 2019; Zhang & Sabuncu, 2018). So it is quite difficult to obtain $\theta^*$ in practice by minimizing $\mathcal{L}_{\text{rob}}(\theta, \tilde{\mathcal{D}})$ over $\theta$. ROBOT prevents this issue by using bilevel optimization. Refer to Remark 1 for more discussion.

**Assumption 2.** *The mapping $\hat{\theta}(\boldsymbol{T})$ is injective, i.e., $\hat{\theta}(\boldsymbol{T}_1) \neq \hat{\theta}(\boldsymbol{T}_2)$ if $\boldsymbol{T}_1 \neq \boldsymbol{T}_2$.*

Then we present the identifiability result given infinite samples as follows:

Table 1: A comparison between MGEO methods with ROBOT. MGEO methods contain anchor-based (Liu & Tao, 2015; Patrini et al., 2017) and anchor-free (Li et al., 2021; Zhang et al., 2021) methods. *The consistency considered here refers to the setting where we do not assume the noisy posterior is perfectly estimated since posterior estimation errors are inevitable in practice.

| | No anchor point assumption | Identifiablility | Consistency* | Finite sample generalization | Error under $g_{\epsilon,\boldsymbol{x}'}$ |
|---|---|---|---|---|---|
| MGEO(Anchor-based) | ✗ | ✓ | ✗Col 1 | ✗ | $\Omega(1)$ (Prop 1) |
| MGEO(Anchor-free) | ✓ | ✓ | ✗(Col 1) | ✗ | $\Omega(1)$ (Prop 1) |
| ROBOT | ✓ | ✓(Thm 1) | ✓(Thm 2) | ✓(Thm 2) | $O(1/n)$ (Prop 2) |

**Theorem 1** (Identifiability). *Suppose Condition 1 -3 (in Appendix A.2 and A.3) and Assumption 1- 2 holds, our framework in equation 7 can uniquely identify $\boldsymbol{T}^*$ with infinite noisy samples, i.e.,*
$$\mathcal{L}_{\mathrm{rob},\tilde{\mathcal{D}}}(\theta(\boldsymbol{T}^*)) < \mathcal{L}_{\mathrm{rob}}(\theta(\boldsymbol{T}), \hat{\mathcal{D}}), \textit{for all } \boldsymbol{T} \neq \boldsymbol{T}^*.$$

See Appendix A.5 for the proof. Theorem 1 shows ROBOT can uniquely learn $\boldsymbol{T}^*$ based on infinite noisy samples. Notably, we neither require the existence of anchor points, nor assume that the noisy posteriors are perfectly estimated, which distinguishes ROBOT from MGEO based works Li et al. (2021); Zhang et al. (2021). We further present the finite sample generalization results as follows:

**Theorem 2** (Finite Sample Generalization and Consistency). *Suppose we have $\tilde{\mathcal{D}}_v(n)$ with $n$ samples. Assume the loss function is bounded by $\ell(\cdot, \cdot) \leq M$. Let the function class be $\mathcal{F} := \{\ell(f_{\hat{\theta}(\boldsymbol{T})}(\cdot); \cdot) : \mathcal{X} \times \mathcal{Y} \to \mathbb{R}^+, \forall \boldsymbol{T} \in \mathcal{T}\}$. Fix any $\epsilon > 0$, assume we obtain a $\epsilon$-approximated solution $\hat{\boldsymbol{T}}$ that satisfies $\mathcal{L}_{\mathrm{rob}}(\hat{\theta}(\hat{\boldsymbol{T}}), \tilde{\mathcal{D}}_v(n)) \leq \mathcal{L}_{\mathrm{rob}}(\hat{\theta}(\boldsymbol{T}), \tilde{\mathcal{D}}_v(n)) + \epsilon, \forall \boldsymbol{T} \in \mathcal{T}$. Let $\mathcal{N}(\epsilon, \mathcal{F}, \|\cdot\|_\infty)$ be the $\epsilon$-cover of $\mathcal{F}$. Then with probability at least $1 - \delta$,*

$$\mathcal{L}_{\mathrm{rob}}(\hat{\theta}(\hat{\boldsymbol{T}}), \tilde{\mathcal{D}}) \leq \inf_{\boldsymbol{T} \in \mathcal{T}} \mathcal{L}_{\mathrm{rob}}(\hat{\theta}(\boldsymbol{T}), \tilde{\mathcal{D}}) + 2\epsilon + M\sqrt{\frac{2\ln(2\mathcal{N}(\epsilon, \mathcal{F}, \|\cdot\|_\infty)/\delta)}{n}}. \tag{8}$$

*Further, if equation 7 is well solved ($\epsilon \to 0$), then $\hat{\boldsymbol{T}} \xrightarrow{p} \boldsymbol{T}^*$ as $n \to \infty$.*

The full proof of Theorem 2 is included in Appendix A.6. We also add additional results on the convergence of $\boldsymbol{T}$-estimation error in Appendix A.11. Due to the instability of MGEO methods, the finite sample guarantees have been missing in MGEO methods. The consistency of our method is a direct consequence of finite sample property. Notably, the consistency of existing works (e.g., Theorem 2 of Zhang et al. (2021)) requires that the noisy posterior is perfectly estimated, which does not hold in a more realistic settings as we consider. We then analyze the stability of ROBOT under the inaccurate noisy posterior $g_{\epsilon,\boldsymbol{x}'}$ as defined in equation 4.

**Proposition 2** (Stability). *Consider we obtain an inaccurate posterior $g_{\epsilon,\boldsymbol{x}'}(\cdot)$ as defined in equation 4, the $\boldsymbol{T}$-estimation error of ROBOT upper bounded as: $\sup_\epsilon \|\boldsymbol{T}_{\epsilon,\boldsymbol{x}'}^{\mathrm{ROBOT}} - \boldsymbol{T}\| \leq O(1/n)$.*

See Appendix A.7 for a proof. Comparing Proposition 2 with Proposition 1, we can see that ROBOT achieves an $O(1/n)$ robustness to the posterior estimation error in the case where MEGO leads to an constant error. Finally, we summarize the comparison between ROBOT with MGEO in Table 1.

## 4 EXPERIMENTS

In this section, we conduct extensive experiments to demonstrate the effectiveness of ROBOT. Our method demonstrates superior performances compared with other state of the art approaches base on loss correction. In particular, we try two robust losses for the outer loop loss function of ROBOT, namely MAE (Ghosh et al., 2017) and RCE (Wang et al., 2019) losses.

**Benchmark Datasets.** We experiment our proposed method on three synthetic datasets: MNIST, CIFAR10 and CIFAR100. In addition, we also conduct experiments on three real world datasets, namely CIFAR10-N, CIFAR100-N and Clothing1M. For synthetic dataset experiments, we run experiments with two types of commonly used noise generation processes: symmetric and pair-flip noises. The experiments were repeated for 5 times, we report both the mean and standard deviation of our results. The results of the baseline approaches are derived from Li et al. (2021). For more details about the datasets and noise generation, please refer to the appendix.

**Baseline Methods, Network Architectures and Training.** We compare our ROBOT with the following baselines: Decoupling (Malach & Shalev-Shwartz, 2017), Co-teaching Han et al. (2018), T-Revision Xia et al. (2019), MentorNet (Jiang et al., 2018), Forward (Patrini et al., 2017),MAE

Table 2: Test accuracy of experiments on MNIST, CIFAR10 and CIFAR100 with different noise types and noise ratios. Our method significantly outperforms the counterparts by a large margin across all experiment settings. Notably, the superiority of our method becomes more evident under challenging scenarios, such as CIFAR100 dataset with flip noise.

| Datasets | MNIST | | CIFAR10 | | CIFAR100 | |
|---|---|---|---|---|---|---|
| | Uniform 20% | Uniform 50% | Uniform 20% | Uniform 50% | Uniform 20% | Uniform 50% |
| Decoupling | 97.04±0.06 | 94.58±0.08 | 77.32±0.35 | 54.07±0.46 | 41.92±0.49 | 22.63±0.44 |
| MentorNet | 97.21±0.06 | 95.56±0.15 | 81.35±0.23 | 73.47±0.15 | 42.88±0.41 | 32.66±0.40 |
| Co-teaching | 97.07±0.10 | 95.20±0.23 | 82.27±0.07 | 75.55±0.07 | 48.48±0.66 | 36.77±0.52 |
| Forward | 98.60±0.19 | 97.77±0.16 | 85.20±0.80 | 74.82±0.78 | 54.90±0.74 | 41.85±0.71 |
| T-Revision | 98.72±0.10 | 98.23±0.10 | 87.95±0.36 | 80.01±0.62 | 62.72±0.69 | 49.12±0.22 |
| MAE | 97.13±0.13 | 95.30±0.15 | 83.28±0.39 | 82.13±0.75 | 55.67±1.03 | 43.53±1.11 |
| RCE | 97.02±0.19 | 95.56±0.30 | 85.63±0.64 | 82.79±0.96 | 53.21±1.10 | 44.24±1.31 |
| DMI | 98.70±0.02 | 98.12±0.21 | 87.54±0.20 | 82.68±0.21 | 62.65±0.39 | 52.42±0.64 |
| Dual T | 98.43±0.05 | 98.15±0.12 | 88.35±0.33 | 82.54±0.19 | 62.16±0.58 | 52.49±0.37 |
| TVD | 98.56±0.09 | 98.11±0.16 | 88.89±0.21 | 83.21±0.13 | 63.52±0.40 | 52.54±1.33 |
| VolMinNet | 98.74±0.08 | 98.23±0.16 | 89.58±0.26 | 83.37±0.25 | 64.94±0.40 | 53.89±1.26 |
| ROBOT (MAE) | **99.01±0.03** | **98.50±0.09** | **92.13±0.07** | **88.75±0.10** | **72.52±0.15** | **64.69±0.32** |
| ROBOT (RCE) | **99.05±0.06** | **98.54±0.14** | **92.09±0.04** | **88.45±0.15** | **73.03±0.12** | **65.11±0.53** |
| Datasets | MNIST | | CIFAR10 | | CIFAR100 | |
| | Flip 20% | Flip 45% | Flip 20% | Flip 45% | Flip 20% | Flip 45% |
| Decoupling | 96.93±0.07 | 94.34±0.54 | 77.13±0.30 | 53.71±0.99 | 40.12±0.26 | 27.97±0.12 |
| MentorNet | 96.89±0.04 | 91.98±0.46 | 77.42±0.00 | 61.03±0.20 | 39.22 ±0.47 | 26.48±0.37 |
| Co-teaching | 97.00±0.06 | 96.25±0.01 | 80.65±0.20 | 73.02 ±0.23 | 42.79±0.79 | 27.97±0.20 |
| Forward | 98.84±0.10 | 95.06±2.61 | 88.21±0.48 | 77.44±6.89 | 56.12±0.54 | 36.88±2.32 |
| T-Revision | 98.89±0.08 | 84.56±8.18 | 90.33±0.52 | 78.94±2.58 | 64.33±0.49 | 41.55±0.95 |
| MAE | 97.43±0.20 | 95.45±0.56 | 79.30±0.13 | 57.13±5.61 | 52.15±1.22 | 39.86±1.50 |
| RCE | 97.10±0.18 | 94.60±0.54 | 82.53±0.35 | 59.65±3.23 | 53.10±1.33 | 40.02±1.47 |
| DMI | 98.84±0.09 | 97.92±0.76 | 89.89±0.45 | 73.15±7.31 | 59.56±0.73 | 38.17±2.02 |
| Dual T | 98.86±0.04 | 96.71±0.02 | 89.77 ±0.25 | 76.53 ±2.51 | 67.21±0.43 | 47.60±0.43 |
| TVD | 98.98±0.04 | 99.12±0.09 | 90.01±0.23 | 88.15±0.10 | 67.65±0.89 | 56.53±0.16 |
| VolMinNet | 99.01±0.07 | 99.00±0.07 | 90.37±0.30 | 88.54±0.21 | 68.45±0.69 | 58.90±0.89 |
| ROBOT (MAE) | **99.39±0.05** | **99.19±0.10** | **93.70±0.07** | **92.52±0.16** | **75.55±0.39** | **70.20±0.33** |
| ROBOT (RCE) | **99.34±0.02** | **99.23±0.10** | **93.62±0.03** | **92.44±0.23** | **75.79±0.51** | **70.17±0.40** |

loss (Ghosh et al., 2017), RCE loss (Wang et al., 2019), DMI loss (Xu et al., 2019),DUAL-T Yao et al. (2020),Total Variation Regularization (Zhang et al., 2021) and VolMinNet (Li et al., 2021). We follow the experiment setting as in Li et al. (2021) and report the baseline results given in their paper. We point out that, similar to Xia et al. (2019; 2021); Li et al. (2021); Wei et al. (2021), the aggregation methods such as SELF (Nguyen et al., 2019) and DivideMix (Li et al., 2020) are not compared in our paper, because such methods incorporate tricks such as semi-supervised training, which can be sensitive to the choice of hyper-parameters. We use the same network architectures as in Li et al. (2021). The experimental details are included in Appendix B.3.

**Experiments on Synthetic Label Noise**. We compare our method with other approaches on commonly used datasets with synthetic label noise as described in Section 4. The results in 2 shows that ROBOT consistently outperforms baselines by a large margin across all datasets and types of label noise. Remarkably, the advantage of our method becomes more evident when the task gets more challenging. For instance, our method outperforms the previous SOTA $T$ estimation approach VolMinNet Li et al. (2021) by 11.22% and 11.27% in test accuracy for CIFAR100 with 50% Uniform noise and 45% pairflip noise, respectively.

**Estimation Error of Transition Matrix**. We compare the $T$-estimation error with other approaches on a variety of datasets with different settings. Note that with synthetic label noise, we have the ground truth $T^*$, and therefore able to calculate the estimation error. The results in Table 3 show that ROBOT achieves lower $T$-estimation error than MGEO-based methods. Note that the comparison between ROBOT and the two-stage methods supports the argument in Remark 1.

**Experiments on Real World Label Noise**. To furthur verify the ability of our method to handle label noise learning, we conduct experiments with datasets containing real-world label noise. Specifically, we showcase the performance of our method on CIFAR10-N, CIFAR100-N (in Table 4) and Clothing1M (in Table 5) datasets. Note that in this paper, we mostly focus on the estimation of transition matrix $T$ that is robust to outliers, therefore, we mainly compare with approaches based on loss correction for fairness. We can observe that our method outperforms other baselines by a noticeable margin, which verifies its ability to handle real world label noise.

Table 3: We compare the noise transition matrix estimation errors between various methods across multiple datasets. Note that Two-stage methods correspond to the alternative approach mentioned in Remark 1. We can see that ROBOT consistently achieves the lowest $T$ estimation error.

| Datasets | MNIST | | CIFAR10 | | CIFAR100 | |
|---|---|---|---|---|---|---|
| | Uniform 20% | Uniform 50% | Uniform 20% | Uniform 50% | Uniform 20% | Uniform 50% |
| Forward | 0.33±0.09 | 0.56±0.10 | 0.39±0.02 | 1.05±0.21 | 0.50±0.10 | 1.11±0.25 |
| T-Revision | 0.17±0.04 | 0.16±0.05 | 0.23±0.09 | 0.30±0.10 | 0.46±0.09 | 0.61±0.12 |
| Two-stage (MAE) | 0.17±0.10 | 0.19±0.18 | 0.43±0.20 | 0.32±0.11 | 0.48±0.15 | 0.70±0.23 |
| Two-stage (RCE) | 0.16±0.10 | 0.21±0.23 | 0.35±0.18 | 0.40±0.12 | 0.49±0.19 | 0.61±0.19 |
| Dual T | 0.13±0.04 | 0.25±0.04 | 0.30±0.05 | 0.77±0.24 | 0.48±0.14 | 1.02±0.25 |
| VolMinNet | 0.07±0.02 | 0.09±0.03 | 0.21±0.06 | 0.12±0.05 | 0.47±0.10 | 0.50±0.13 |
| ROBOT (MAE) | **0.03±0.01** | **0.03±0.01** | **0.10±0.01** | **0.10±0.03** | **0.18±0.06** | **0.32±0.06** |
| ROBOT (RCE) | **0.03±0.01** | **0.03±0.02** | **0.07±0.03** | **0.09±0.02** | **0.20±0.05** | **0.31±0.07** |
| Datasets | MNIST | | CIFAR10 | | CIFAR100 | |
| | Flip 20% | Flip 45% | Flip 20% | Flip 45% | Flip 20% | Flip 45% |
| Forward | 0.31±0.07 | 0.77±0.15 | 0.45±0.09 | 0.97±0.12 | 0.48±0.12 | 0.96±0.15 |
| T-Revision | 0.12±0.02 | 0.25±0.04 | 0.20±0.10 | 0.29±0.12 | 0.45±0.13 | 0.92±0.34 |
| Two-stage (MAE) | 0.16±0.10 | 0.22±0.09 | 0.20±0.12 | 0.32±0.11 | 0.50±0.23 | 0.90±0.44 |
| Two-stage (RCE) | 0.13±0.08 | 0.24±0.10 | 0.19±0.15 | 0.40±0.12 | 0.47±0.30 | 0.95±0.50 |
| Dual T | 0.10±0.02 | 0.45±0.09 | 0.32 ±0.07 | 0.75 ±0.20 | 0.45±0.12 | 0.98±0.21 |
| VolMinNet | 0.04±0.01 | 0.04±0.01 | 0.12±0.03 | 0.13±0.04 | 0.13±0.04 | 0.25±0.07 |
| ROBOT (MAE) | **0.02±0.01** | **0.03±0.01** | **0.09±0.02** | **0.11±0.01** | **0.11±0.03** | **0.22±0.03** |
| ROBOT (RCE) | **0.02±0.01** | **0.03±0.01** | **0.07±0.01** | **0.10±0.03** | **0.12±0.05** | **0.19±0.05** |

Table 4: Test accuracy of experiments on CIFAR10-N and CIFAR100-N with different noise types. For fair comparison, we only compare against the approaches that are designed based on transition matrices. Our method consistently outperforms the counterparts across all experiment settings.

| Datasets | CIFAR10-N | | | | | CIFAR100-N |
|---|---|---|---|---|---|---|
| | Aggregate | Random 1 | Random 2 | Random 3 | Worst | Noisy |
| CE | 87.77±0.38 | 85.02±0.65 | 86.46±1.79 | 85.16±0.61 | 77.69±1.55 | 50.50±0.66 |
| MAE | 85.63±0.55 | 84.38±0.34 | 84.64±0.30 | 85.05±0.29 | 77.34±1.56 | 51.24±1.35 |
| RCE | 85.02±0.67 | 84.80±0.49 | 84.23±0.50 | 84.03±0.33 | 77.43±0.97 | 51.80±1.44 |
| GCE | 87.85±0.70 | 87.61±0.28 | 87.70±0.56 | 87.58±0.29 | 80.66±0.53 | 56.73±0.30 |
| Backward-T | 88.13±0.29 | 87.14±0.34 | 86.28±0.80 | 86.86±0.41 | 77.61±1.05 | 57.14±0.94 |
| Forward-T | 88.24±0.22 | 86.88±0.50 | 86.14±0.21 | 87.04±0.35 | 79.49±0.46 | 57.01±1.03 |
| T-Revision | 88.52±0.17 | 88.33±0.32 | 87.71±1.02 | 87.79±0.67 | 80.48±1.20 | 51.55±0.31 |
| TVD | 88.96±0.15 | 88.41±0.13 | 88.35±0.11 | 88.01±0.40 | 80.15±0.25 | 56.89±0.32 |
| VolMinNet | 89.70±0.12 | 88.30±0.12 | 88.27±0.09 | 88.19±0.41 | 80.53±0.20 | 57.80±0.31 |
| Peer loss | 90.75±0.25 | 89.06±0.11 | 88.76±0.19 | 88.57±0.19 | 82.53±0.52 | 57.59±0.61 |
| ROBOT (MAE) | **91.14±0.12** | **90.35±0.14** | **90.30±0.16** | **90.34±0.16** | **84.08±0.22** | **61.14±0.29** |
| ROBOT (RCE) | **91.35±0.03** | **90.46±0.18** | **90.37±0.15** | **90.31±0.21** | **84.05±0.33** | **61.25±0.26** |

Table 5: Test accuracy of experiments on Clothing1M. We only adopt noisy data during training.

| Decoupling | MentorNet | Co-teaching | Forward | Dual T | T-Revision |
|---|---|---|---|---|---|
| 54.53 | 56.79 | 60.15 | 69.91 | 71.49 | 70.97 |
| DMI | PTD | TVD | VolminNet | ROBOT (MAE) | ROBOT (RCE) |
| 70.97 | 70.12 | 71.67 | 72.42 | 72.64 | **72.70** |

## 5 CONCLUSION

In this paper, we investigate the problem of learning statistically consistent models under label noise by estimating $T$. We first propose the framework MGEO to unify the existing $T$-estimation methods. Then we provide both theoretical and experimental results to show that MGEO methods are sensitive to the error in noisy posterior estimation. To overcome the limitation of MGEO, we further propose ROBOT, which enjoys superior theoretical properties and shows strong empirical performance.

## REPRODUCIBILITY STATEMENT

The experiments in the paper are all conducted using public datasets. The hyperparameters and network choices for the experiments are elaborated in Section 4 and Appendix B. We submit the source code with the ICLR submission. The code will be make public upon acceptance of the paper.

For the theory part, the assumptions and full proof are included in Section 3.2, Appendix A.4, A.5, A.6 and A.6.

## ACKNOWLEDGEMENTS

BH was supported by NSFC Young Scientists Fund No. 62006202, Guangdong Basic and Applied Basic Research Foundation No. 2022A1515011652, RGC Early Career Scheme No. 22200720, CAAI-Huawei MindSpore Open Fund and HKBU CSD Departmental Incentive Grant. XBX was supported by Australian Research Council Project DE-190101473 and Google PhD Fellowship. TLL was partially supported by Australian Research Council Projects IC-190100031, LP-220100527, DP-220102121, and FT-220100318.

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

## A    PROOFS

### A.1    INTRODUCTION OF EXISTING METHODS

In Section 2, we show that the framework MEGO takes anchor-based and anchor-free (the minimum volume method) methods as special cases. In this section, we briefly introduce the anchor-based method (Patrini et al., 2017; Liu & Tao, 2015) and the minimum volume method Li et al. (2021) for completeness. Here we assume that the fitted posterior $\tilde{g}(\boldsymbol{x})$ matches the noisy posterior, e.g., $\tilde{g}(\boldsymbol{x}) = P(\tilde{Y}|\boldsymbol{X} = \boldsymbol{x})$, for all $\boldsymbol{x}$.

**The anchor-based method.** For each class $j \in [K]$, anchor-based methods assumes that there exists an anchor point $\bar{\boldsymbol{x}}^j$ with $P(Y|\boldsymbol{X} = \bar{\boldsymbol{x}}^j) = \boldsymbol{e}^j$. Anchor based method first find the most confident sample for each class:

$$\bar{\boldsymbol{x}}^j = \arg\max_{\boldsymbol{x}} \tilde{g}(\boldsymbol{x})[j], \tag{9}$$

where $\tilde{g}(\boldsymbol{x})[j]$ is the probability of the class $j$ of the sample $\boldsymbol{x}$ given by $\tilde{g}(\boldsymbol{x})$. By the anchor point assumption, we have

$$T_{\cdot,j} = \boldsymbol{T}\boldsymbol{e}^j = \boldsymbol{T}P(Y|\boldsymbol{X} = \bar{\boldsymbol{x}}^j) = \tilde{g}(\bar{\boldsymbol{x}}^j), \tag{10}$$

where $T_{\cdot,j}$ is the $j$th column of $\boldsymbol{T}$. By repeating equation 9 and equation 10 for each class, anchor points methods obtain the estimation of the whole $\boldsymbol{T}$.

**The anchor-based method (the minimum volume method).**    Since the posterior $\tilde{g}(\boldsymbol{x}) = \boldsymbol{T}P(Y|X = \boldsymbol{x})$, then $\tilde{g}(\boldsymbol{x})$ is enclosed in the convex hull formed by the columns of $\boldsymbol{T}$. However, there are still infinite number of $\boldsymbol{T}$ whose conv($\boldsymbol{T}$) encloses all the samples. When the samples are sufficiently scattered, Li et al. (2021) shows that $\boldsymbol{T}^*$ is the one with the minimum volume. So the solve the following problem:

$$\min_{\boldsymbol{T} \in \mathcal{T}} \text{vol}(\boldsymbol{T}) \tag{11}$$

$$s.t.\ \boldsymbol{T}f_\theta(\boldsymbol{x}) = \tilde{g}(\boldsymbol{x}), \forall \boldsymbol{x}, \tag{12}$$

where vol($\boldsymbol{T}$) is the volume of $\boldsymbol{T}$.

### A.2    THE CONDITIONS FOR THE ROBUST LOSSES

The robust losses can work well (i.e., equation 6 holds) when the noise ratio is not too large. As shown in the Theorem 1 of Wang et al. (2019), we need the noise ratio to satisfy the following condition:

**Condition 1** (Restate of Theorem 1 in Wang et al. (2019))*. equation 6 holds 1) under symmetric or uniform label noise if noise rate $\eta < 1/K$, where $K$ is the class number; 2) under asymmetric or class-dependent label noise when noise rate $\eta_{yk} \leq 1 - \eta_y$, with $\sum_{k \neq y} \eta_{yk} = \eta_y$.*

A.3    THE CONDITIONS FOR THE CONSISTENCY OF FORWARD CORRECTION METHOD

The conditions of forward correction is presented in the Section 4.2 of Patrini et al. (2017). We discuss these conditions briefly in this section for completeness. Recall the risk on dataset $\mathcal{D}(n)$ is $\mathcal{L}(\theta, \mathcal{D}(n)) := \frac{1}{n} \sum_{(\boldsymbol{x},y) \in \mathcal{D}(n)} [\ell(f_\theta(\boldsymbol{x}), y)]$ where $\ell$ is the loss function. We discuss the cross entropy loss as $\ell$ in this section since we focus on classification tasks. Here we consider the loss function $\ell$ is endowed with a *link function* $\phi$: $\Delta^{K-1} \rightarrow \mathbb{R}^K$, where $K$ is the class number. In the case of cross entropy, the softmax is the inverse link function, i.e.,

$$f_\theta(x) = \phi^{-1}(h_\theta(x))$$
$$\ell(f_\theta(x), y) = \ell_\phi(h_\theta(x), y).$$

The first condition is that the function space parameterized by $\theta$ is large enough, i.e.,

**Condition 2.** *The function class is sufficiently large such that there exists a $\theta^*$ and $f_{\theta^*}(x) = \mathbb{P}[y|x]$.*

Notably, by the universal approximation property of neural networks (Scarselli & Tsoi, 1998), Condition 2 can be satisfied by using a deep and wide neural network. In this work, we use Lenet for MNIST dataset, ResNet18 for CIFAR10 and CIFAR10-N, ResNet34 for for CIFAR100 and CIFAR100-N, and ResNet50 for Clothing1M (the same with existing works). The second condition condition is that the composite losses are proper as follows:

**Condition 3.** *Suppose Condition 2 holds, the composite loss is proper as follows.*

$$\arg\min_{h_\theta} \ell_\phi(h_\theta(x), y) = \phi(p(y|x)).$$

Notably, cross entropy and square loss are examples of proper composite losses.

Theorem 2 of Patrini et al. (2017) shows that if Condition 2 and 3 hold, minimizing the forward correction loss on the noisy data leads to the optimal function which minimizes the loss on the clean data.

A.4    ASSUMPTIONS AND PROOF OF PROPOSITION 1

**Assumption 3.** *Assume the follows are true:*

   *(a)  There exists label noise that is constant level , i.e.,*

$$\max_{i \in [K]} (1 - T_{ii}) \geq C \geq \Omega(1).$$

   *(b)  Let $\mu$ denotes the Lebesgue measure, for any $\boldsymbol{T}_1, \boldsymbol{T}_2 \in \mathcal{V}$ if Conv($\boldsymbol{T}_1$) $\subsetneq$ Conv($\boldsymbol{T}_2$), then*

$$\mathcal{M}(\boldsymbol{T}_2) - \mathcal{M}(\boldsymbol{T}_1) \geq \Omega(\mu(\text{Conv}(\boldsymbol{T}_2) \setminus \text{Conv}(\boldsymbol{T}_2))).$$

   *(c)  If $\mathcal{M}(\boldsymbol{T}_1) > \mathcal{M}(\boldsymbol{T}_2)$, then*

$$\|\boldsymbol{T}_1 - \boldsymbol{T}_2\|_F \geq \Omega(\mathcal{M}(\boldsymbol{T}_1) - \mathcal{M}(\boldsymbol{T}_2)).$$

   *(d)  $\tilde{g}(\boldsymbol{x}')$ does not lies in the boundary of Conv($\boldsymbol{T}$) such that $\tilde{g}(\boldsymbol{x}') = \sum_{i=1}^K \alpha_i \boldsymbol{t}_i$ and $0 < \alpha_i < 1, \forall i$.*

The first assumption is natural for noisy data problems that the noise ratio is larger than 0. The intuition for the Assumption 3(b) is that the difference in metric should be the same order with the difference in the measure of the convex hulls. For example, if Conv($\boldsymbol{T}_1$) is larger than Conv($\boldsymbol{T}_2$) one by a constant level in Lebesgue measure, the metric of $\boldsymbol{T}_1$ is also larger than that of $\boldsymbol{T}_2$ by a constant level. The Assumption 3(c) requires that if two matrix $\boldsymbol{T}_1$ and $\boldsymbol{T}_2$ is different in terms of the measure $\mathcal{M}$, there should be difference between their elements, which is captured by the Frobenius norm. One can easily check these assumptions holds for anchor points and minimum volume methods illustrated in Example 1 and 2. The Assumption 3(d) is also a mild assumption because it holds almost surely.

*Proof.* Denote $\boldsymbol{v}_k$ as a vector with $1$ at the $k$th index and $0$ at the other index, i.e.,

$$v_{k,i} = \begin{cases} 1, & \text{if } i = k, \\ 0, & \text{otherwise} \end{cases}.$$

Let

$$\bar{\boldsymbol{T}}^{\text{MGEO}} := \underset{\boldsymbol{V} \in \mathcal{V}}{\arg\min} \, \mathcal{M}(\boldsymbol{V}), \text{ s.t. } \mathcal{G}(n) \subset \text{Conv}(\boldsymbol{V}),$$

and

$$\bar{\boldsymbol{T}}^{\text{MGEO}}_{\epsilon, \boldsymbol{x}'} := \underset{\boldsymbol{V} \in \mathcal{V}}{\arg\min} \, \mathcal{M}(\boldsymbol{V}), \text{ s.t. } \mathcal{G}(n, \epsilon) \subset \text{Conv}(\boldsymbol{V}).$$

Under the assumptions of anchor points Patrini et al. (2017) or sufficiently scattered Li et al. (2021),

$$\bar{\boldsymbol{T}}^{\text{MGEO}} = \boldsymbol{T}. \tag{13}$$

Take $j = \arg\min_i T_{ii}$ and $\epsilon_j = -g(\boldsymbol{x}') + \boldsymbol{e}_j$. In this case, we claim

$$\bar{\boldsymbol{T}}^{\text{MGEO}}_{\epsilon, \boldsymbol{x}'} = \boldsymbol{T}_{e_j}, \tag{14}$$

where $\boldsymbol{T}_{e_j} := [\boldsymbol{t}_1, \boldsymbol{t}_2 ... \boldsymbol{e}_j ... \boldsymbol{t}_K]$. This is because 1) we have $\mathcal{G}(n, \epsilon) \subset \text{Conv}(\boldsymbol{T}_{e_j})$ because $\mathcal{G}(n) \subset \text{Conv}(\boldsymbol{T}_{e_j})$ and $g_{\epsilon, \boldsymbol{x}'}(\boldsymbol{x}') \in \text{Conv}(\boldsymbol{T}_{e_j})$; 2) One can easily check $\text{Conv}([\boldsymbol{t}_1, \boldsymbol{t}_2 ... \boldsymbol{t}_j ... \boldsymbol{t}_K, \boldsymbol{e}_j]) = \text{Conv}([\boldsymbol{t}_1, \boldsymbol{t}_2 ..., \boldsymbol{e}_j ... \boldsymbol{t}_K]) = \text{Conv}(\boldsymbol{T}_{e_j})$ and $\text{Conv}([\boldsymbol{t}_1, \boldsymbol{t}_2 ... \boldsymbol{t}_j ... \boldsymbol{t}_K, \boldsymbol{e}_j])$ is the smallest convex hull that contains both $\text{Conv}([\boldsymbol{t}_1, \boldsymbol{t}_2 ... \boldsymbol{t}_j ... \boldsymbol{t}_K])$ and $\{\boldsymbol{e}_j\}$ in terms of measure $\mathcal{M}$. Further $\text{Conv}([\boldsymbol{t}_1, \boldsymbol{t}_2 ... \boldsymbol{t}_j ... \boldsymbol{t}_K, \boldsymbol{e}_j])$ is the smallest convex hull containing $\mathcal{G}(n)$. With Assumption 3 (d) we know $\text{Conv}(\boldsymbol{T}_{e_j})$ is the smallest convex hull that contains $\mathcal{G}(n, \epsilon)$.

Further, given any $\alpha_i \in (T_{ii}, 1]$, we can see $\boldsymbol{v}_{e_i} = \alpha_i \boldsymbol{e}_i + \sum_{[K] \backslash \{i\}} \alpha_j \boldsymbol{t}_j \notin \text{Conv}(\boldsymbol{T})$ for any $\sum_{j \in [K] \backslash \{i\}} \alpha_j = 1 - \alpha_i$ and $\alpha_j \geq 0$. This is because for all $\boldsymbol{v} \in \text{Conv}(\boldsymbol{T})$, $v_i \leq T_{ii}$ but the $i$th element of $\boldsymbol{v}_{e_i}$ is larger than $T_{ii}$. On the other hand, one can easily see that $\text{Conv}(\boldsymbol{T}) \subset \text{Conv}(\boldsymbol{T}_{e_i})$. So we have

$$\mu(\text{Conv}(\boldsymbol{T}_{e_1}) \backslash \text{Conv}(\boldsymbol{T}))$$
$$\geq \mu(\{\boldsymbol{v} | \boldsymbol{v} = \alpha_i \boldsymbol{e}_i + \sum_{j \in [K] \backslash \{i\}} \alpha_j \boldsymbol{t}_j, \alpha_i \in (T_{ii}, 1], \sum_{j \in [K] \backslash \{i\}} \alpha_j = 1 - \alpha_i, \alpha_j \geq 0\})$$
$$= \Omega(1 - T_{ii}), \tag{15}$$

Then we have

$$\sup_\epsilon \|\bar{\boldsymbol{T}}^{\text{MGEO}}_{\epsilon, \boldsymbol{x}'} - \bar{\boldsymbol{T}}^{\text{MGEO}}\|_{\text{F}}$$
$$\geq \sup_\epsilon \|\bar{\boldsymbol{T}}^{\text{MGEO}}_{\epsilon_j, \boldsymbol{x}'} - \bar{\boldsymbol{T}}^{\text{MGEO}}\|_{\text{F}}$$
$$= \|\boldsymbol{T}_{e_j} - \boldsymbol{T}\|_{\text{F}}$$
$$\geq \Omega(\mathcal{M}(\boldsymbol{T}_{e_j}) - \mathcal{M}(\boldsymbol{T}))$$
$$\geq \Omega\left(\mu(\text{Conv}(\boldsymbol{T}_{e_j}) \backslash \text{Conv}(\boldsymbol{T}))\right)$$
$$\geq \Omega(1 - T_{jj})$$
$$\geq \Omega(1).$$

The first inequality is due to taking $\epsilon$ as $\epsilon_j$; the first inequality is due to equation 13 and equation 14; the second inequality is due to Assumption 3 (c); the third inequality is due to Assumption 3 (b) and the last inequality is due to equation 15.

$\square$

## A.5 PROOF OF THEOREM 1

*Proof.* First there exists a $\boldsymbol{T}$ that can induce $\theta^*$ in the inner loop. This is due the consistency of forward/backward correction: when $\boldsymbol{T} = \boldsymbol{T}^*$, $\theta(\boldsymbol{T}) \to \theta^*$ as $n \to \infty$. Then by Assumption 1, we know $\mathcal{L}_{\text{rob}}(f_{\theta(\boldsymbol{T}^*)}, \mathcal{D}) < \mathcal{L}_{\text{rob}}(f_\theta, \mathcal{D}), \forall \theta \neq \theta^*$. As verified in Ghosh et al. (2017); Wang et al. (2019); Xu et al. (2019), we have $\mathcal{L}_{\text{rob}}(f_\theta, \tilde{\mathcal{D}}) = \mathcal{L}_{\text{rob}}(f_\theta, \mathcal{D}) + c$ for any $\theta$, where $c$ is a fixed constant. We then have $\mathcal{L}_{\text{rob}}(f_{\theta(\boldsymbol{T}^*)}, \tilde{\mathcal{D}}) < \mathcal{L}_{\text{rob}}(f_\theta, \tilde{\mathcal{D}}), \forall \theta \neq \theta^*$. Finally, we further have $\mathcal{L}_{\text{rob}}(\theta(\boldsymbol{T}^*)) < \mathcal{L}_{\text{rob}}(\theta(\boldsymbol{T})), \forall \boldsymbol{T} \neq \boldsymbol{T}^*$ by Assumption 2. $\square$

## A.6 Proof of Theorem 2

*Proof.* Recall the definition of $\epsilon$ covering, we can find a $\boldsymbol{T}_i$ for $\hat{\boldsymbol{T}}$ in the covering set such that $|\ell(\hat{\theta}(\boldsymbol{T}_i); \boldsymbol{X}, Y) - \ell(\hat{\theta}(\hat{\boldsymbol{T}}); \boldsymbol{X}, Y)| \leq \epsilon, \forall (\boldsymbol{X}, Y) \in \mathcal{X} \times \mathcal{Y}$, then we have:

$$
\begin{aligned}
&\mathcal{L}_{\text{rob}}(\hat{\theta}(\hat{\boldsymbol{T}}), \tilde{\mathcal{D}}) \\
\leq &\mathcal{L}_{\text{rob}}(\hat{\theta}(\boldsymbol{T}_i), \tilde{\mathcal{D}}) + \epsilon \\
\leq &\mathcal{L}_{\text{rob}}(\hat{\theta}(\boldsymbol{T}_i), \tilde{\mathcal{D}}_v^n) + M\sqrt{\frac{\ln(2\mathcal{N}(\epsilon, \mathcal{F}, \|\cdot\|_\infty)/\delta)}{2n}} + \epsilon \\
\leq &\mathcal{L}_{\text{rob}}(\hat{\theta}(\hat{\boldsymbol{T}}), \tilde{\mathcal{D}}_v^n) + M\sqrt{\frac{\ln(2\mathcal{N}(\epsilon, \mathcal{F}, \|\cdot\|_\infty)/\delta)}{2n}} + 2\epsilon \\
\leq &\mathcal{L}_{\text{rob}}(\hat{\theta}(\boldsymbol{T}), \tilde{\mathcal{D}}_v^n) + M\sqrt{\frac{\ln(2\mathcal{N}(\epsilon, \mathcal{F}, \|\cdot\|_\infty)/\delta)}{2n}} + 3\epsilon \\
\leq &\mathcal{L}_{\text{rob}}(\hat{\theta}(\boldsymbol{T}), \tilde{\mathcal{D}}) + M\sqrt{\frac{\ln(2/\delta)}{2n}} + M\sqrt{\frac{\ln(2\mathcal{N}(\epsilon, \mathcal{F}, \|\cdot\|_\infty)/\delta)}{2n}} + 3\epsilon \\
\leq &\mathcal{L}_{\text{rob}}(\hat{\theta}(\boldsymbol{T}), \tilde{\mathcal{D}}) + M\sqrt{\frac{2\ln(2\mathcal{N}(\epsilon, \mathcal{F}, \|\cdot\|_\infty)/\delta)}{n}} + 3\epsilon
\end{aligned}
$$

The first and third inequalities are by the definition of $\epsilon$ covering, the second inequality is to apply Hoeffding inequality on all elements of the covering sets, the forth inequality is due to $\hat{\boldsymbol{T}}$ is $\epsilon$-approximated solution on the dataset $\tilde{\mathcal{D}}_v^n$, the fifth inequality is to apply Hoeffding inequality on $\boldsymbol{T}$ and the last inequality in because $\mathcal{N}(\epsilon, \mathcal{F}, \|\cdot\|_\infty) > 1$. □

## A.7 Proof of Proposition 2

**Assumption 4.** *Assume the first and second derivatives involved have norm bounded above and the inverse matrices have all positive eigenvalues :*

*(a)* $\sigma_{\min}\left(\frac{\partial \mathcal{L}_{\text{rob}}(f_\theta, \tilde{\mathcal{D}}^n)}{\partial \theta} \frac{\partial^2 \theta_\epsilon(\boldsymbol{T})}{\partial \boldsymbol{T}^2}\right) \geq Q_1 > 0$,

*(b)* $\sigma_{\min}\left(\frac{\partial^2 \mathcal{L}(\boldsymbol{T} f_\theta, \tilde{\mathcal{D}}^n)}{\partial \theta \partial \theta^\top}\right) \geq Q_2 > 0$,

*(c)* $\|\frac{\partial \theta(\boldsymbol{T})}{\partial \boldsymbol{T}}\|_2 \leq Q_3$,

*(d)* $\|\frac{\partial \mathcal{L}_{\text{rob}}(f_\theta, \tilde{\mathcal{D}}^n)}{\partial \theta}\|_2 \leq Q_5$,

*(e)* $\|\frac{\partial^2 \mathcal{L}(\boldsymbol{T} f_\theta, \tilde{\mathcal{D}}^n)}{\partial \theta \partial \boldsymbol{T}^\top}\|_2 \leq Q_6$

*(f)* $\|\frac{\partial k(\theta_\epsilon, \boldsymbol{T}, \boldsymbol{x}', y', \epsilon)}{\partial \theta}\|_2 \leq Q_7$,

*(g)* $\|\frac{\partial^2 k(\theta, \boldsymbol{T}, \boldsymbol{x}', y', \epsilon)}{\partial \theta \partial \theta^\top}\|_2 \leq Q_8$.

We imediately have

$$
\left\|\left(\frac{\partial \mathcal{L}_{\text{rob}}(f_\theta, \tilde{\mathcal{D}}^n)}{\partial \theta} \frac{\partial^2 \theta_\epsilon(\boldsymbol{T})}{\partial \boldsymbol{T}^2}\right)^{-1}\right\|_F \leq d \left\|\left(\frac{\partial \mathcal{L}_{\text{rob}}(f_\theta, \tilde{\mathcal{D}}^n)}{\partial \theta} \frac{\partial^2 \theta_\epsilon(\boldsymbol{T})}{\partial \boldsymbol{T}^2}\right)^{-1}\right\|_2 \leq \frac{d}{Q_1}
$$

**Lemma 1** (Cauchy, Implicit Function Theorem, Theorem 1 of Lorraine et al. (2020)). *If for some* $(\theta, \boldsymbol{T})$ *such that* $\frac{\partial \mathcal{L}(\boldsymbol{T} f_\theta, \tilde{\mathcal{D}})}{\partial \theta}\big|_{\theta', \boldsymbol{T}'} = 0$ *and regularity conditions are satisfied, then surrounding* $(\theta', \boldsymbol{T}')$ *there exists a function* $\theta^*(\boldsymbol{T})$ *such that* $\frac{\partial \mathcal{L}(\boldsymbol{T} f_\theta, \tilde{\mathcal{D}})}{\partial \theta}\big|_{\theta^*(\boldsymbol{T}), \boldsymbol{T}} = 0$ *and we have*

$$
\frac{\partial \theta^*(\boldsymbol{T})}{\partial \boldsymbol{T}}\bigg|_{\boldsymbol{T}'} = \left[\frac{\partial^2 \mathcal{L}(\boldsymbol{T} f_\theta, \tilde{\mathcal{D}})}{\partial \theta \partial \theta^\top}\right]^{-1} \times \frac{\partial^2 \mathcal{L}(\boldsymbol{T} f_\theta, \tilde{\mathcal{D}})}{\partial \theta \partial \boldsymbol{T}^\top}\bigg|_{\theta^*(\boldsymbol{T}'), \boldsymbol{T}'}
$$

We consider that the noisy posterior perfectly fit the dataset $\tilde{\mathcal{D}}^n(\epsilon) := \tilde{\mathcal{D}}^{n-1} \cup (\boldsymbol{x}', y' + \epsilon)$, which is equivalent to the inaccurate poster $g_{\epsilon, \boldsymbol{x}'}$.

$$\theta_\epsilon(\boldsymbol{T}) = \arg\min_\theta \frac{n-1}{n} \mathcal{L}(\boldsymbol{T}f_\theta, \tilde{\mathcal{D}}^{n-1}) + \frac{1}{n} \mathcal{L}(\boldsymbol{T}f_\theta, (\boldsymbol{x}', y' + \epsilon))$$

$$= \arg\min_\theta \mathcal{L}(\boldsymbol{T}f_\theta, \tilde{\mathcal{D}}^n) + \frac{1}{n} \left( \ell(\boldsymbol{T}f_\theta(\boldsymbol{x}'), y' + \epsilon)) - \ell(\boldsymbol{T}f_\theta(\boldsymbol{x}'), y') \right)$$

$$= \arg\min_\theta \mathcal{L}(\boldsymbol{T}f_\theta, \tilde{\mathcal{D}}^n) + \frac{1}{n} k(\theta, \boldsymbol{T}, \boldsymbol{x}', y', \epsilon)$$

Because $\frac{\partial \mathcal{L}(\boldsymbol{T}f_\theta, \tilde{\mathcal{D}}^n)}{\partial \theta} = 0$, we have

$$-\frac{1}{n} \frac{\partial k(\theta_\epsilon, \boldsymbol{T}, \boldsymbol{x}', y', \epsilon)}{\partial \theta}$$

$$= \frac{\partial \mathcal{L}(\boldsymbol{T}f_{\theta_\epsilon}, \tilde{\mathcal{D}}^n)}{\partial \theta}$$

$$= \frac{\partial \mathcal{L}(\boldsymbol{T}f_{\theta_\epsilon}, \tilde{\mathcal{D}}^n)}{\partial \theta} - \frac{\partial \mathcal{L}(\boldsymbol{T}f_\theta, \tilde{\mathcal{D}}^n)}{\partial \theta}$$

$$= \frac{\partial^2 \mathcal{L}(\boldsymbol{T}f_\theta, \tilde{\mathcal{D}}^n)}{\partial \theta \partial \theta^\top} (\theta - \theta_\epsilon) + o(\theta - \theta_\epsilon)$$

Then

$$\frac{\partial \theta_\epsilon(\boldsymbol{T})}{\partial \boldsymbol{T}} - \frac{\partial \theta(\boldsymbol{T})}{\partial \boldsymbol{T}} = \left[ \frac{\partial^2 \mathcal{L}(\boldsymbol{T}f_\theta, \tilde{\mathcal{D}}^n(\epsilon))}{\partial \theta \partial \theta^\top} \right]^{-1} \times \frac{\partial^2 \mathcal{L}(\boldsymbol{T}f_\theta, \tilde{\mathcal{D}}^n(\epsilon))}{\partial \theta \partial \boldsymbol{T}^\top} - \frac{\partial \theta(\boldsymbol{T})}{\partial \boldsymbol{T}}$$

$$= \left[ \frac{\partial^2 \mathcal{L}(\boldsymbol{T}f_\theta, \tilde{\mathcal{D}}^n)}{\partial \theta \partial \theta^\top} + \frac{1}{n} \frac{\partial^2 k(\theta, \boldsymbol{T}, \boldsymbol{x}', y', \epsilon)}{\partial \theta \partial \theta^\top} \right]^{-1} \times \left[ \frac{\partial^2 \mathcal{L}(\boldsymbol{T}f_\theta, \tilde{\mathcal{D}}^n)}{\partial \theta \partial \boldsymbol{T}^\top} + \frac{1}{n} \frac{\partial^2 k(\theta, \boldsymbol{T}, \boldsymbol{x}', y', \epsilon)}{\partial \theta \partial \boldsymbol{T}^\top} \right] - \frac{\partial \theta(\boldsymbol{T})}{\partial \boldsymbol{T}}$$

$$= \frac{1}{n} \left( \frac{\partial^2 \mathcal{L}(\boldsymbol{T}f_\theta, \tilde{\mathcal{D}}^n)}{\partial \theta \partial \theta^\top} \right)^{-2} \frac{\partial^2 k(\theta, \boldsymbol{T}, \boldsymbol{x}', y', \epsilon)}{\partial \theta \partial \theta^\top} \frac{\partial^2 \mathcal{L}(\boldsymbol{T}f_\theta, \tilde{\mathcal{D}}^n)}{\partial \theta \partial \boldsymbol{T}^\top}$$

$$+ \frac{1}{n} \left( \frac{\partial^2 \mathcal{L}(\boldsymbol{T}f_\theta, \tilde{\mathcal{D}}^n)}{\partial \theta \partial \theta^\top} \right)^{-1} \frac{\partial^2 \mathcal{L}(\boldsymbol{T}f_\theta, \tilde{\mathcal{D}}^n)}{\partial \theta \partial \boldsymbol{T}^\top} + o(1/n)$$

$$= \frac{1}{n} \boldsymbol{J} + o(1/n),$$

where

$$\boldsymbol{J} = \left( \frac{\partial^2 \mathcal{L}(\boldsymbol{T}f_\theta, \tilde{\mathcal{D}}^n)}{\partial \theta \partial \theta^\top} \right)^{-2} \frac{\partial^2 k(\theta, \boldsymbol{T}, \boldsymbol{x}', y', \epsilon)}{\partial \theta \partial \theta^\top} \frac{\partial^2 \mathcal{L}(\boldsymbol{T}f_\theta, \tilde{\mathcal{D}}^n)}{\partial \theta \partial \boldsymbol{T}^\top} + \left( \frac{\partial^2 \mathcal{L}(\boldsymbol{T}f_\theta, \tilde{\mathcal{D}}^n)}{\partial \theta \partial \theta^\top} \right)^{-1} \frac{\partial^2 \mathcal{L}(\boldsymbol{T}f_\theta, \tilde{\mathcal{D}}^n)}{\partial \theta \partial \boldsymbol{T}^\top}$$

By Assumption 4, we have

$$\|\boldsymbol{J}\|_2 \leq \frac{Q_6 Q_8}{Q_2^2} + \frac{Q_6}{Q_2}$$

Then we have

$$(\theta - \theta_\epsilon) = \frac{1}{n} \left( \frac{\partial^2 \mathcal{L}(\boldsymbol{T}f_\theta, \tilde{\mathcal{D}}^n)}{\partial \theta \partial \theta^\top} \right)^{-1} \frac{\partial k(\theta_\epsilon, \boldsymbol{T}, \boldsymbol{x}', y', \epsilon)}{\partial \theta}$$

Then we have the following by omitting the higher order terms:

$$\frac{\partial \mathcal{L}_{\text{rob}}(f_{\theta_\epsilon}, \tilde{\mathcal{D}}^n(\epsilon))}{\partial \theta}$$

$$= \frac{\partial \mathcal{L}_{\text{rob}}(f_{\theta_\epsilon}, \tilde{\mathcal{D}}^n)}{\partial \theta} + \frac{1}{n} \frac{\partial k(\theta, \boldsymbol{T}, \boldsymbol{x}', y', \epsilon)}{\partial \theta}$$

$$= \frac{\partial \mathcal{L}_{\text{rob}}(f_\theta, \tilde{\mathcal{D}}^n)}{\partial \theta} + \frac{\partial^2 \mathcal{L}_{\text{rob}}(f_\theta, \tilde{\mathcal{D}}^n)}{\partial^2 \theta} (\theta_\epsilon - \theta) + \frac{1}{n} \frac{\partial k(\theta, \boldsymbol{T}, \boldsymbol{x}', y', \epsilon)}{\partial \theta} + o(1/n)$$

And further:

$$\frac{\partial \theta_\epsilon(\boldsymbol{T}_\epsilon)}{\partial \boldsymbol{T}}$$

$$=\frac{\partial \theta_\epsilon(\boldsymbol{T}_\epsilon)}{\partial \boldsymbol{T}} - \frac{\partial \theta_\epsilon(\boldsymbol{T})}{\partial \boldsymbol{T}} + \frac{\partial \theta_\epsilon(\boldsymbol{T})}{\partial \boldsymbol{T}} - \frac{\partial \theta(\boldsymbol{T})}{\partial \boldsymbol{T}} + \frac{\partial \theta(\boldsymbol{T})}{\partial \boldsymbol{T}}$$

$$=\frac{\partial^2 \theta_\epsilon(\boldsymbol{T})}{\partial \boldsymbol{T}^2}(\boldsymbol{T}_\epsilon - \boldsymbol{T}) + \frac{1}{n}\boldsymbol{J} + \frac{\partial \theta(\boldsymbol{T})}{\partial \boldsymbol{T}} + o(1/n)$$

On the other side, we have

$$0 =\frac{\partial \mathcal{L}_{\text{rob}}(f_{\theta(\boldsymbol{T}_\epsilon)}, \tilde{\mathcal{D}}^n(\epsilon))}{\partial \boldsymbol{T}}$$

$$=\frac{\partial \mathcal{L}_{\text{rob}}(f_{\theta_\epsilon}, \tilde{\mathcal{D}}^n(\epsilon))}{\partial \theta}\frac{\partial \theta_\epsilon(\boldsymbol{T}_\epsilon)}{\partial \boldsymbol{T}}$$

$$=\frac{\partial \mathcal{L}_{\text{rob}}(f_{\theta_\epsilon}, \tilde{\mathcal{D}}^n(\epsilon))}{\partial \theta}\frac{\partial \theta_\epsilon(\boldsymbol{T}_\epsilon)}{\partial \boldsymbol{T}}$$

$$=\left(\frac{\partial \mathcal{L}_{\text{rob}}(f_{\theta_\epsilon}, \tilde{\mathcal{D}}^n)}{\partial \theta} + \frac{1}{n}\frac{\partial \left(\ell(\boldsymbol{T}f_\theta(\boldsymbol{x}'), y' + \epsilon)) - \ell(\boldsymbol{T}f_\theta(\boldsymbol{x}'), y')\right)}{\partial \theta}\right)\frac{\partial \theta_\epsilon(\boldsymbol{T}_\epsilon)}{\partial \boldsymbol{T}}$$

$$=\left(\frac{\partial \mathcal{L}_{\text{rob}}(f_\theta, \tilde{\mathcal{D}}^n)}{\partial \theta} + \frac{\partial^2 \mathcal{L}_{\text{rob}}(f_\theta, \tilde{\mathcal{D}}^n)}{\partial \theta \partial \theta^\top}(\theta_\epsilon - \theta) + \frac{1}{n}\frac{\partial k(\theta, \boldsymbol{T}, \boldsymbol{x}', y', \epsilon)}{\partial \theta}\right)$$

$$\times \left(\frac{\partial \theta(\boldsymbol{T})}{\partial \boldsymbol{T}} + \frac{\partial^2 \theta_\epsilon(\boldsymbol{T})}{\partial \boldsymbol{T}^2}(\boldsymbol{T}_\epsilon - \boldsymbol{T}) + \frac{1}{n}\boldsymbol{J} + o(1/n)\right)$$

Denote

$$\boldsymbol{K} :=\frac{\partial^2 \mathcal{L}_{\text{rob}}(f_\theta, \tilde{\mathcal{D}}^n)}{\partial \theta \partial \theta^\top}\left(\frac{\partial^2 \mathcal{L}(\boldsymbol{T}f_\theta, \tilde{\mathcal{D}}^n)}{\partial \theta \partial \theta^\top}\right)^{-1}\frac{\partial k(\theta_\epsilon, \boldsymbol{T}, \boldsymbol{x}', y', \epsilon)}{\partial \theta} + \boldsymbol{I},$$

With Assumption 4, we have

$$\|\boldsymbol{K}\|_2 \le \frac{Q_5 Q_7}{Q_2} + 1$$

Recall that

$$\frac{\partial \mathcal{L}_{\text{rob}}(f_\theta, \tilde{\mathcal{D}}^n)}{\partial \theta}\frac{\partial \theta(\boldsymbol{T})}{\partial \boldsymbol{T}} = 0,$$

then we have following by omitting high order terms:

$$\frac{1}{n}\frac{\partial \mathcal{L}_{\text{rob}}(f_\theta, \tilde{\mathcal{D}}^n)}{\partial \theta}\boldsymbol{J} + \frac{\partial \mathcal{L}_{\text{rob}}(f_\theta, \tilde{\mathcal{D}}^n)}{\partial \theta}\frac{\partial^2 \theta_\epsilon(\boldsymbol{T})}{\partial \boldsymbol{T}^2}(\boldsymbol{T}_\epsilon - \boldsymbol{T}) = -\frac{1}{n}\boldsymbol{K}\frac{\partial \theta(\boldsymbol{T})}{\partial \boldsymbol{T}}.$$

Then finally we have

$$\|(\boldsymbol{T}_\epsilon - \boldsymbol{T})\|_{\text{F}}$$

$$\le K\|(\boldsymbol{T}_\epsilon - \boldsymbol{T})\|_2$$

$$=\frac{K}{n}\left\|\left(\frac{\partial \mathcal{L}_{\text{rob}}(f_\theta, \tilde{\mathcal{D}}^n)}{\partial \theta}\frac{\partial^2 \theta_\epsilon(\boldsymbol{T})}{\partial \boldsymbol{T}^2}\right)^{-1}\left(\boldsymbol{K}\frac{\partial \theta(\boldsymbol{T})}{\partial \boldsymbol{T}} + \frac{\partial \mathcal{L}_{\text{rob}}(f_\theta, \tilde{\mathcal{D}}^n)}{\partial \theta}\boldsymbol{J}\right)\right\|_2$$

$$\le\frac{K}{n}\left\|\left(\frac{\partial \mathcal{L}_{\text{rob}}(f_\theta, \tilde{\mathcal{D}}^n)}{\partial \theta}\frac{\partial^2 \theta_\epsilon(\boldsymbol{T})}{\partial \boldsymbol{T}^2}\right)^{-1}\right\|_2\left\|\left(\boldsymbol{K}\frac{\partial \theta(\boldsymbol{T})}{\partial \boldsymbol{T}} + \frac{\partial \mathcal{L}_{\text{rob}}(f_\theta, \tilde{\mathcal{D}}^n)}{\partial \theta}\boldsymbol{J}\right)\right\|_2$$

$$\le\frac{K}{n}\frac{1}{Q_1}\left(\left\|\boldsymbol{K}\frac{\partial \theta(\boldsymbol{T})}{\partial \boldsymbol{T}}\right\|_2 + \left\|\frac{\partial \mathcal{L}_{\text{rob}}(f_\theta, \tilde{\mathcal{D}}^n)}{\partial \theta}\boldsymbol{J}\right\|_2\right)$$

$$\le\frac{K}{n}\frac{1}{Q_1}\left((\frac{Q_5 Q_7}{Q_2} + 1)Q_3 + Q_5(\frac{Q_6 Q_8}{Q_2^2} + \frac{Q_6}{Q_2})\right)$$

$$=\frac{Q}{n}.$$

where $Q = \frac{1}{Q_1}(\frac{Q_5 Q_7}{Q_2} + 1)Q_3 + Q_5(\frac{Q_6 Q_8}{Q_2^2} + \frac{Q_6}{Q_2})$ and $Q_1 - Q_8$ are specified in Assumption 4.

A.8   ALGORITHM OF ROBOT

Usually the neural network parameters $\theta$ is of high dimension, and the number of samples in the training dataset $\tilde{\mathcal{D}}_{tr}$ is huge, which makes exactly solving the inner problem infeasible. Therefore, in practice, we solve the inner problem and the outer problem alternatively at each update step to alleviate the computational burden Shu et al. (2019); Ren et al. (2018). We provide detailed explanation of how our ROBOT works in practice.

**Forward Correction with Noise Transition Matrix $T$.**   With the noise transition matrix $T$, the procedure of one-step update in forward correction method in the inner loop can be formulated as:

$$\theta^{t+1} = \theta^t - \eta \frac{1}{n} \sum_{i=1}^{n} \frac{\partial l_{ce}(\boldsymbol{T} f_\theta(x_i), y_i)}{\partial \theta} \tag{16}$$

where $n$ is the number of training samples in a mini-batch, $\eta$ is the learning rate.

**Approximately Solving the Inner Problem.**   Due to the high time complexity for exactly solving $\hat{\theta}(\boldsymbol{T}) = \arg\min_\theta \mathcal{L}(\boldsymbol{T} f_\theta, \tilde{\mathcal{D}}_{tr})$, we use one-step update to approximate it, which is widely used in previous literature and shown to be effective Shu et al. (2019); Ren et al. (2018). The approximation at the $\theta^t$ can be formulated as follows:

$$\tilde{\theta}(\boldsymbol{T}) = \theta^t - \eta \frac{1}{n} \sum_{i=1}^{n} \frac{\partial l_{ce}(\boldsymbol{T} f_\theta(x_i), y_i)}{\partial \theta} \tag{17}$$

**Update $T$ In the Outer Loop.**   With the approximate solution from the inner loop, we establish a mapping from $\boldsymbol{T}$ to $\tilde{\theta}(\boldsymbol{T})$, with which we can calculate the gradient of the outer loss w.r.t $\boldsymbol{T}$. Therefore, the update in the outer loop can be formulated as:

$$\boldsymbol{T}^{t+1} = \boldsymbol{T}^t - \alpha \frac{1}{m} \sum_{i=1}^{m} \frac{\partial l_{rob}(f_{\tilde{\theta}(\boldsymbol{T})}(x_i), y_i)}{\partial \boldsymbol{T}} \tag{18}$$

---

**Algorithm 1** Algorithm of ROBOT

---

**Require:** Noisy training dataset $\tilde{\mathcal{D}}_{tr}$, noisy validation dataset $\tilde{\mathcal{D}}_v$, batch size $n$ and $m$, outer update frequency $K$, total iterations $E$.
1: Initialize transition matrix $T$ with an identity matrix.
2: **for** training iteration $t = 1, 2 \ldots E$ **do**
3:   $\{x_v, y_v\} \leftarrow \text{SampleBatch}(\tilde{\mathcal{D}}_v, m)$
4:   $\{x_t, y_t\} \leftarrow \text{SampleBatch}(\tilde{\mathcal{D}}_t, m)$
5:   **if** $t\ /\ K$ is 0 (# update $\boldsymbol{T}$ once each $K$ iteration.) **then**
6:     Solve the approximation for inner problem as 17
7:     Update $\boldsymbol{T}$ by Equation 18
8:   **end if**
9:   Update $\theta$ by Equation 16
10: **end for**
   **Output:** Optimized $\boldsymbol{T}^*$ and $\theta^*$.

---

**Discussion about the Algorithm.**   In practice, the noisy training dataset $\tilde{\mathcal{D}}_{tr}$ and the noisy validation dataset $\tilde{\mathcal{D}}_v$ can be the same, which achieves similar performance. Our method is able to scale because the one-step approximation used in our implementation. Besides, the frequency of the outer loop updates can be set lower for better efficiency. With these approximation technique, our framework can be efficiently trained on large-scale datasets, e.g., Cloth1M with 1 million samples. The overall computational cost is about 1.6 times the cost for regular training on Cloth1M.

A.9   CONVERGENCE OF ROBOT

The convergence of the bilevel optimization using approximated solution of inner loop was first established in Pedregosa (2016). We restate it here for completeness.

**Theorem 3** (Convergence, Theorem 3.3 of Pedregosa (2016)). *Suppose $\mathcal{L}_{\mathrm{rob}}(f_{\hat{\theta}(\boldsymbol{T})}; \tilde{\mathcal{D}})$ is smooth w.r.t. $\boldsymbol{T}$, $\mathcal{L}(\boldsymbol{T} f_\theta; \tilde{\mathcal{D}})$ is $\beta$-smooth and $\alpha$-strongly convex w.r.t. $\theta$. We solve the inner loop by unrolling $J$ steps, choose learning rate in the outer loop as $\eta_\tau = 1/\sqrt{\tau}$, then we arrive at a approximately stationary point as follows after $R$ steps:*

$$\mathbb{E}\left[\sum_{\tau=1}^{R} \frac{\eta_\tau \|\nabla_{\boldsymbol{T}} \mathcal{L}_{\mathrm{rob}}(f_{\hat{\theta}(\boldsymbol{T})}; \tilde{\mathcal{D}})\|_2^2}{\sum_{\tau=1}^{R} \eta_\tau}\right] \leq \tilde{O}\left(\epsilon + \frac{\epsilon^2 + 1}{\sqrt{R}}\right), \tag{19}$$

*where $\tilde{O}$ absorbs constants and logarithmic terms and $\epsilon = (1 - \alpha/\beta)^J$.*

### A.10 IMPLEMENTATION OF ROBUT LOSSES

We provide the details about the robust loss functions used in ROBOT.

**Reverse Cross Entropy Loss (RCE)**

$$\ell_{\mathrm{rce}}(f(x, \boldsymbol{\theta}), y) = -\sum_{k=1}^{K} f_k(x, \boldsymbol{\theta}) \log q(k|x) \tag{20}$$

**Mean Absolute Error (MAE)**

$$\ell_{\mathrm{mae}}(f(x, \boldsymbol{\theta}), y) = -\sum_{k=1}^{K} \|f_k(x) - q(k|x)\|_1 \tag{21}$$

We denote the ground-truth distribution over labels by $q(k|x)$, and $\sum_{k=1}^{K} q(k|x) = 1$. Given the ground-truth label is $y$, then $q(y|x) = 1$ and $q(k|x) = 0$ for all $k \neq y$. For the RCE loss, we approximate $\log(0)$ to constant 10.

### A.11 CONVERGENCE OF $T$

In Theorem 2, we provide the uniform convergence of ROBOT in terms of the outer loss. We are also able to direct bound the deviation of $\hat{\boldsymbol{T}}$ from $\boldsymbol{T}^*$ as follows:

**Corollary 2.** *Define $\epsilon, \mathcal{N}(\epsilon, \mathcal{F}, \|\cdot\|_\infty), \hat{\boldsymbol{T}}$ as the same with Theorem 2. Assume $\mathcal{L}_{\mathrm{rob}}(\hat{\theta}(\boldsymbol{T}), \tilde{\mathcal{D}})$ is $\gamma$-strongly-convex w.r.t. $\boldsymbol{T}$, then with probability at least $1 - \delta$, we have*

$$\|\hat{\boldsymbol{T}} - \boldsymbol{T}^*\|_2 \leq \sqrt{\frac{2\epsilon + M\sqrt{\frac{2\ln(2\mathcal{N}(\epsilon, \mathcal{F}, \|\cdot\|_\infty)/\delta)}{n}}}{\gamma}}. \tag{22}$$

*Proof.* By Theorem 2 and the fact $\boldsymbol{T}^* = \arg\min_{\boldsymbol{T}} \mathcal{L}_{\mathrm{rob}}(\hat{\theta}(\boldsymbol{T}), \tilde{\mathcal{D}})$, we have with probability at least $1 - \delta$ the following holds

$$\mathcal{L}_{\mathrm{rob}}(\hat{\theta}(\hat{\boldsymbol{T}}), \tilde{\mathcal{D}}) \leq \mathcal{L}_{\mathrm{rob}}(\hat{\theta}(\boldsymbol{T}^*), \tilde{\mathcal{D}}) + 2\epsilon + M\sqrt{\frac{2\ln(2\mathcal{N}(\epsilon, \mathcal{F}, \|\cdot\|_\infty)/\delta)}{n}}. \tag{23}$$

By the strong convexity, we have

$$\mathcal{L}_{\mathrm{rob}}(\hat{\theta}(\hat{\boldsymbol{T}}), \tilde{\mathcal{D}}) \geq \mathcal{L}_{\mathrm{rob}}(\hat{\theta}(\hat{\boldsymbol{T}}^*), \tilde{\mathcal{D}}) + \nabla_{\boldsymbol{T}} \mathcal{L}_{\mathrm{rob}}(\hat{\theta}(\hat{\boldsymbol{T}}^*), \tilde{\mathcal{D}})(\hat{\boldsymbol{T}} - \boldsymbol{T}^*) + \frac{\gamma}{2}\|\hat{\boldsymbol{T}} - \boldsymbol{T}^*\|_2^2 \tag{24}$$

By the optimality of $\boldsymbol{T}^*$, we have

$$\nabla_{\boldsymbol{T}} \mathcal{L}_{\mathrm{rob}}(\hat{\theta}(\boldsymbol{T}), \tilde{\mathcal{D}}) = 0. \tag{25}$$

Combining equation 23-25, we know that with probability at least $1 - \delta$, the following holds:

$$\frac{\gamma}{2}\|\hat{\boldsymbol{T}} - \boldsymbol{T}^*\|_2^2 \leq 2\epsilon + M\sqrt{\frac{2\ln(2\mathcal{N}(\epsilon, \mathcal{F}, \|\cdot\|_\infty)/\delta)}{n}}. \tag{26}$$

We finish the proof by rearrangement. $\qquad\square$

# B EXPERIMENTAL DETAILS

## B.1 PLOT DETAILS

We choose 3 classes from MNIST (namely, class 2, 4 and 6), and apply uniform noise with noise rate 0.4. For ease of illustration, we randomly sample 100 points for each class. We first fit a 3-laryer MLP $h(\boldsymbol{x})$ on the noisy dataset. Then we the Minimum Volume method (Li et al., 2021) to find the minimum $\boldsymbol{T}$ whose conv($\boldsymbol{T}$) encloses all samples. We then train ROBOT on the same setting. To make a fair comparison with minimum volme method, we use $h(\boldsymbol{x})$ as the soft noisy label for ROBOT to ensure that the noisy posterior for ROBOT cannot be more accurate than the minimum volumn method.

## B.2 DATASET DETAILS

CIFAR10-N and CIFAR100-N Wei et al. (2021) is a recently proposed dataset, which was created with the Amazon Mechanical Turk (M-Turk) by posting the CIFAR-10 and CIFAR-100 datasets as the annotation Human Intelligence Tasks (HITs). The human annotations are then used as labels for the training data. Clothing1M is a dataset proposed in Xiao et al. (2015). The dataset contains 1 million images with noisy labels obtained from the web. We follow the same setting as in Li et al. (2021) and only use the noisy dataset to jointly train both the neural network and the noise transition matrix $\boldsymbol{T}$.

## B.3 EXPERIMENTAL DETAILS

Specifically, we train Lenet with 5 layers for MNIST, SGD with batch size 128, weight decay $10^{-3}$ momentum 0.9 and learning rate $10^{-2}$ is used to optimize the neural network parameters, while Adam with learning rate $10^{-2}$ and batch size 128 is used to optimize $T$ in the outer loop; for CIFAR10, we experiment on ResNet18, trained using SGD with batch size 128, weight decay $5 \times 10^{-4}$ momentum 0.9 and learning rate $5 \times 10^{-2}$; for CIFAR100, we experiment on ResNet34, trained using SGD with batch size 128, weight decay $1 \times 10^{-3}$, momentum 0.9 and learning rate $5 \times 10^{-2}$, for both CIFAR10 and CIFAR100, Adam with learning rate $5 \times 10^{-3}$ and batch size 256 is used din the outer loop to train T; for CIFAR10-N and CIFAR100-N, ResNet34 is training using SGD with learning rate 0.1, momentum 0.9 and weight decay $5 \times 10^{-4}$, following the official hyper-parameters, while T is optimized using Adam with learning rate $5 \times 10^{-3}$; for Clothing1M, we follow Li et al. (2021) to finetune an ImageNet pre-trained ResNet50 using SGD with learning rate $2 \times 10^{-3}$, momentum 0.9 and weight decay $1 \times 10^{-3}$, batch size is set to 32.

## B.4 BILEVEL FORMULATION LEADS TO EASIER OPTIMIZATION

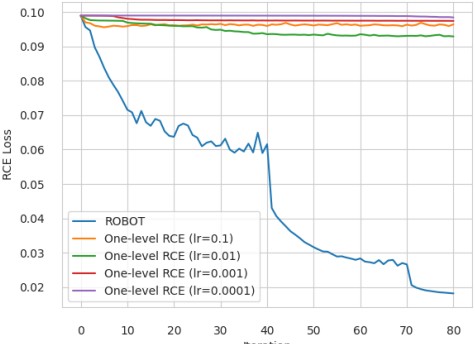

Figure 2: Comparison of RCE loss values on the training dataset achieved by one-level training (directly optimizing the robust loss) and our ROBOT. We can see that directly training the network with RCE leads to difficulty in optimization. On the other hand, ROBOT decreases the training loss quickly since as ROBOT transforms the optimization to a much smaller space (transform the optimization from the space of neural networks to the space of $\boldsymbol{T}$).

We conduct the following two experiments on CIFAR100 to verify that the bilevel formulation in ROBOT leads to easier optimization of the robust loss function: 1) directly using the reverse cross entropy loss to train the network (referred as one-level method in the following discussion); 2) execute the bilevel procedure in our ROBOT (for fair comparison, we use the training dataset for both the inner loop and outer loop in equation 7). In Figure 2, we can see that direct optimizing the robust loss (one level) can hardly decrease the training loss, which is consistent with the findings in Zhang & Sabuncu (2018). On the other hand, the robust training loss optimized as the outer loss of ROBOT decreases rapidly.

**Implementation Details** In ROBOT, we set $K = 1$ for the algorithm described in Appendix A.8 for ROBOT, which iterate between inner and outer loop by performing one step gradient descent each time, in the inner loop, SGD with momentum 0.9 and learning rate 0.1 is used; in the outer loop, Adam with learning rate 0.001 is used. For directly training the model using the reverse cross entropy loss, we have tried different learning rate ($\{0.1, 0.01, 0.001, 0.0001\}$) with Adam optimizer. The learning rate is decayed at 40 and 70 epochs in all runs. For each experiment, we record the value of the reverse cross entropy loss on the training dataset, which is the training loss in the one-level case and is the outer loss in ROBOT, respectively. The results are shown in Figure 2. Each iteration of ROBOT needs two times of gradient evaluations, one for the inner loop and one for the outer loop. Each iteration of direct optimizing robust loss (one-level) needs a single time of gradient evaluation.

### B.5 Experiments w/wo Separate Validation Set

In the implementation of equation 7 described in Appendix A.8, we split the noisy dataset into a training and validation dataset. The training dataset is used in the inner loop and validation set used for the outer loop. Actually we can also use the same noisy dataset for both the inner and outer loops (with out splitting it into a training and validation set). We conduct experiments on CIFAR10 dataset with 20% and 50% uniform noise to compare these two schemes. Table 6 reports the test performances and estimation error rate. The outer objective adopts RCE loss. Other configurations follow the same settings as the main experiment. Table 6 shows that the two schemes lead to similar performance.

Table 6: Comparison between whether using a separate validation set in the outer loop.

| Method | Noise rate=0.2 | Noise rate=0.5 |
|---|---|---|
| Separate Validation | 92.13±0.07 | 88.75 ±0.10 |
| No Separate Validation | 92.09±0.09 | 88.63 ±0.12 |

## C Related Work

In this section we categorize previous noisy label learning approaches into two types: heuristic methods and statistically consistent methods, and further provide a brief introduction.

**Heuristic Methods.** Due to the empirical observation that the neural networks tend to learn easy (correct) samples first, and then starts to fit onto the hard (corrupt) samples in the later phase of training, many algorithms are designed based on the training samples' loss values (Han et al., 2018; Wei et al., 2020; Huang et al., 2019; Pleiss et al., 2020; Yao et al., 2021b). Specifically, the samples with small loss values are presumed to be correctly labelled, while those with large loss values are considered corrupted. Even though these methods demonstrate strong empirical results, they typically lack theoretical guarantees and hence make their reliability questionable.

**Loss Correction Methods.** Algorithms belonging to this category attempt to train a statistically consistent classifier under label noise with theoretical guarantees by utilizing the noise transition matrix ($T$) to correct the loss during training. The majority of previous methods rely on the anchor point assumption, which means there is at least one data belonging each specific class with probability one (Patrini et al., 2017; Xia et al., 2020; Liu & Tao, 2015; Scott, 2015; Scott et al., 2013; Yao et al., 2020; Zhu et al., 2021; Wu et al., 2021; Xia et al., 2022; Li et al., 2022b). However, the

anchor point assumptions are unfavorable (Xia et al., 2019). Recently, (Li et al., 2021; Zhang et al., 2021) make the attempt to estimate $T$ without relying on the anchor point assumption. Even though promising results are achieved, we point out that these methods are prone to inaccurate posteriors estimation (Section 2) and suffer unreliable performances especially when training data is scarce (Section 4). In comparison, our method overcomes the above mentioned issues and demonstrates superior performances.

**Robust Loss Functions.** Several noise-robust loss functions are proposed to train the network (Ghosh et al., 2017; Liu & Guo, 2020; Xu et al., 2019; Wang et al., 2019; Ma et al., 2020; Zhou et al., 2021; Kim et al., 2021), such that when the number of training data approaches infinity, the optimal weights derived from noisy training data is the same as the weights derived from clean data. Despite they learn a robust classifier in theory, they are typically difficult to train the DNNs and result require more hyper-parameter tuning (Wang et al., 2019). Our method utilizes the robust loss functions to optimize the noise transition matrix T instead of the model parameters, the learnt T is then used to correct the loss during training to learn a statistically consistent classifier.

**Bilevel Optimization** Bilevel optimization Sinha et al. (2017) has achieved a lot of successes in recent years, which is able to solve hierarchical decision making processes. Bilevel optimization is adopted in broad areas of research, such as hyper-paramter optimization Lorraine et al. (2020); Maclaurin et al. (2015); Pedregosa (2016); MacKay et al. (2019); Franceschi et al. (2017); Vicol et al. (2021), neural architecture search Pham et al. (2018); Liu et al. (2018); Pham et al. (2018); Shi et al. (2020); Yao et al. (2021a); Gao et al. (2022; 2021); Shi et al. (2021), meta learning Finn et al. (2017); Nichol & Schulman (2018), dataset condensation Wang et al. (2018); Zhao et al. (2020); Cazenavette et al. (2022); Pi et al. (2022) and sample re-weighting Ren et al. (2018); Shu et al. (2019); Zhou et al. (2022a;c).

