# OpenReview forum: "A Holistic View of Label Noise Transition Matrix in Deep Learning and Beyond"
_ICLR.cc/2023/Conference — ICLR 2023 notable top 25%_

### Official Review · Reviewer_FXEG · 2022-10-25

**Confidence:** 3
**Correctness:** 3
**Technical Novelty And Significance:** 3
**Empirical Novelty And Significance:** Not applicable
**Recommendation:** 8

**Clarity, Quality, Novelty And Reproducibility:**

It seems that some description of the forward-correction problem is somewhat misleading (in the introduction and the proof of Theorem 1). The authors repeatedly say in the paper that the forward-correction-based method gives the same theta as the clean loss, but in my understanding, this is not entirely correct. According to [Patrini et al. 2017], what is guaranteed for forward-correction problems is much weaker than that for backward-correction problems, i.e., the minimizer will be the same for both problems if f_theta is allowed to be any functions. However, in practice, f_theta is limited to a certain class, and in that case, the actual solutions of theta can be different. Please correct me if I'm wrong.

The actual optimization procedure for the bilevel problem is not described in the paper. Is the complete computation graph for the lower-level problem constructed and used for the upper-level problem? Or, do the authors simply use an alternative optimization? Although the authors provide the code, some basic explanations regarding this can be discussed in the paper since this is related to the "heaviness" of the training procedure. I have only briefly gone through the code, and the former seems to be the case. How heavy is the computational burden in this regard?

Regarding reproducibility, the authors have provided the code.

**Strength And Weaknesses:**

The main motivation of the paper is valid, i.e., MGEO methods are vulnerable to errors in posterior estimation. The authors provide this with a proposition, and the main reason for this vulnerability is the strict inequality constraint for the "enclosing convex hull" part. The idea for resolving this issue is novel, i.e., combining robust loss functions with the estimation of T in a bilevel formulation. The upper-level problem finds theta (the parameter) based on the robust loss function while the lower-level problem defines theta based on T with a forward-correction-based problem. The authors claim that this kills two birds with one stone in that (i) the difficulty of direct optimization for theta with the robust loss can be alleviated and (ii) the vulnerability of T estimation in MGEO can be bypassed (since now T is determined by the robust loss function through theta). This is quite an interesting approach, and the good performance makes it more convincing.

Although the approach is interesting, it seems some of the positive effects of the proposed method are not fully justified. Among the above two advantages, the benefits of (ii) are clear: Now T is not defined based on the strict convex hull inequality constraint, so it is less sensitive to errors in posterior estimation as also shown in Proposition 2. However, (i) is somewhat unclear. The authors argue that reparametrizing theta with T as in (7) will make the optimization of the robust loss function easier, but the exact mechanism for this is not explained. A clear justification is needed for this part.


**Summary Of The Paper:**

This paper proposes a new method for deep learning with noisy labels. The method is based on the independent noise transition assumption, i.e., the label noise is independent of the input data. Existing methods, either with or without anchors, attempt to find a minimum enclosing convex hull for the noise transition matrix T. However, the authors show that this approach is vulnerable to errors in posterior estimation. To resolve this issue, the authors propose to bypass this problem using a bilevel formulation; Now, we are minimizing a robust loss function that can give the same parameter (theta) for both noisy labels and clean labels, but the parameter itself is defined based on another forward-correction-based optimization problem, where T is the variable. The authors claim that this can also alleviate the difficulty of minimizing the robust loss function directly with theta. The experiments show that the proposed method achieves state-of-the-art results for various benchmark sets.

**Summary Of The Review:**

The main idea for bypassing two difficulties (one for robust loss functions, another for vulnerability of the MGEO problem) at once is novel and interesting. However, the justification that the proposed method can alleviate the former difficulty is not clearly given in the paper. The performance is good, which makes the method convincing.

[After rebuttal] I'm satisfied with the authors' answers and maintain the original score.

---

> ### Author Response · Authors · 2022-11-15
> **Author Response 2/2**
>
> > "*It seems that some description of the forward-correction problem is somewhat misleading (in the introduction and the proof of Theorem 1). The authors repeatedly say in the paper that the forward-correction-based method gives the same theta as the clean loss, but in my understanding, this is not entirely correct. According to [Patrini et al. 2017], what is guaranteed for forward-correction problems is much weaker than that for backward-correction problems, i.e., the minimizer will be the same for both problems if f_theta is allowed to be any functions. However, in practice, f_theta is limited to a certain class, and in that case, the actual solutions of theta can be different.*"
>
> Re: Thanks for pointing out our improper description! We missed the detailed discussion on the conditions for the consistency of forward-correction. Indeed, the forward-correction method needs more strict conditions than the backward, and the sufficiently large function space is one of them. Specifically, we need the space of $f$ to contain $P(y|x)$.  In theory, we can always achieve this condition by using wide and deep neural networks according to the universal approximation property of neural networks [2]. In practice, we use LeNet for MNIST dataset, ResNet18 for CIFAR10 and CIFAR10-N, ResNet34 for CIFAR100 and CIFAR100-N, and ResNet50 for Clothing1M. These choices of networks are consistent with existing works. Notably, [1] shows that these modern neural network architectures are extremely expressive.  We also find that they work out well in experiments. The forward correction method also needs the loss to be proper composite as discussed in [Patrini et al. 2017]. We formally re-state these conditions in Appendix A.3 of the revised version and mention the conditions in the main part of the paper. Thank you very much for pointing this out!
>
> [1] Chiyuan Zhang, et. al., Understanding deep learning (still) requires rethinking generalization.
>
> [2] F Scarselli, et.al., Universal approximation using feedforward neural networks: A survey of some existing methods, and some new results
>
> > "*The actual optimization procedure for the bilevel problem is not described in the paper. Is the complete computation graph for the lower-level problem constructed and used for the upper-level problem? Or, do the authors simply use an alternative optimization?*"
>
> Re: Thank you very much for pointing out this issue! In our implementation, the network does not need to be trained to convergence in each inner iteration. We adopt one-step approximation for the inner problem, which avoids solving the inner problem exactly and does not need to construct the complete computation graph for each step of the outer update. To be specific, we alternatively update $T$ (outer loop) and the network parameters (inner loop). These approximation techniques have been widely used in previous works and demonstrate superior performance [1, 2]. We have included the details of our algorithm in Appendix A.8 and A.10 in the revised version.
>
> [1] Jun shu, et. al., Meta-Weight-Net: Learning an Explicit Mapping For Sample Weighting
>
> [2] Amirreza Shaban, et. al., Truncated Back-propagation for Bilevel Optimization, AISTATS 2019
>
> > "*How heavy is the computational burden in this regard?*"
>
> Re: With this approximation technique mentioned above, our framework can be efficiently trained on large-scale datasets, e.g., Clothing1M with 1 million samples. The overall computational cost is about 1.6 times the cost for regular training on Clothing1M. We also add discussiones on this in Appendix A.8.

---

> > ### Comment · Reviewer_FXEG · 2022-11-16
> > **Thank you for the detailed explanation.**
> >
> > Thank you for elaborating on my concerns. I'm satisfied with the authors' answers regarding these points.

---

> ### Author Response · Authors · 2022-11-15
> **Author Response 1/2**
>
> > "*The authors claim that this kills two birds with one stone in that (i) the difficulty of direct optimization for theta with the robust loss can be alleviated and (ii) ... However, (i) is somewhat unclear. The authors argue that reparametrizing theta with T as in (7) will make the optimization of the robust loss function easier, but the exact mechanism for this is not explained. A clear justification is needed for this part.*"
>
> Re: Thank you for pointing out this important question. In the previous paper version, we mentioned that our Bilevel framework can make the optimization of the robust loss much easier but didn’t provide a detailed explanation of the mechanism and further support. Sorry for missing this important part.
>
> **Experimental Support**. In the revised paper, we add a section in Appendix B.4 to provide experimental results for this claim and add more explanation in Remark 1 in Section 4.1. Specifically, we compare the trend of training robust loss by directly optimizing the robust loss (referred as the one-level method) and optimizing ROBOT. For a fair comparison, we use the same training dataset for the inner and outer loops of ROBOT. We record the training robust loss of the one-level method and the outer loss (also the training robust loss) of ROBOT. The result presented in Appendix B.4 shows that directly optimizing the robust loss cannot decrease the training loss effectively (even if we try different learning rates on it), which is consistent with the findings in [1]. In contrast, ROBOT can significantly reduce training robust loss significantly. (Refer to Appendix B.4 for the implementation details).
>
> **More Explanation**. The difficulty of directly optimizing the robust loss is first discussed [1]. [1] shows that the gradient induced by robust loss is much less effective than the gradient of ordinary cross-entropy loss.  First, let’s us consider  directly minimizing  ${L}_{rob}(f_\theta, D)$ through optimizing over $\theta$. Typically, we use an overparameterized neural network, i.e., ResNet18 for Cifar10, which contains millions of parameters. So $\theta$’s dimension is over a million. We want to optimize $\theta$ to some oracle parameter $\theta^*$. Since $\theta$ is randomly initialized and has a very high dimension, the distance between $\theta^*$ and $\theta$, i.e., $\|\theta - \theta^*\|_1 = O(million)$.
>
> Because the gradient induced by the robust loss is relatively ineffective, it is hard to guide $\theta$ to $\theta^*$ over such a long distance. So in the figure of Appendix B.4, the training robust can not decrease effectively.
>
> Now let’s consider using ROBOT. We are optimizing ${L_{rob}(f_{\theta(T)}, D)}$  over $ T $ in the outer loop. Note that $\theta(T)$ is well solved in the inner loop because cross-entropy is easy to optimize. So the problem now is to find a good $T$ to minimize $L_{rob}(f_{\theta(T)}, D)$. The distance between a randomly initialized $T$ and $T^*$ is $|T - T^*|_1$,  which is in the order $|T - T^*|_1 = O(K^2)$, where $K$ is the number of classes. Even in Cifar100, we have $|T - T^*|_1 = O(10000)$,
> which is much smaller than the distance of $|\theta - \theta^*|_1 = O(million)$. In other words, we can find the optimal solution in a place that is much closer to our initialization.
>
> Thanks again for this important suggestion!
>
> [1] Zhilu Zhang and Mert Sabuncu. Generalized cross entropy loss for training deep neural networks with noisy labels

---

> > ### Comment · Reviewer_FXEG · 2022-11-16
> > **Thank you for the detailed explanation.**
> >
> > Thank you for elaborating on my concerns. The additional justification provided in the paper, especially the additional experiment part, is very good. However, I think the "more explanation" part is somewhat weak. The authors' argument is basically that higher dimensionality will lead to a longer distance to optimum, so reparametrizing the variable into a lower dimensional space makes the optimization easier. This is not always true, and what actually matters is the landscape of the loss function w.r.t. the optimization variable. There are many cases where higher dimensional parametrizations can give better solutions, and sometimes lower dimensional parametrizations can make the problem more difficult. In other words, dimensionality alone is not sufficient to argue the efficiency of parametrization. The authors must either provide a better justification or tone down regarding the advantage of reparametrization (=the reparametrization MAY have better convergence property) and fall back to the empirical justification (Appendix B.4).

---

> > > ### Author Response · Authors · 2022-11-17
> > > **Further Response**
> > >
> > > Thanks for the insightful comment! We agree that the most important factor to accelerate the convergence should be a better landscape of the loss function w.r.t. the optimization variable (e.g., smoothness, convexity). We have adjusted our statement on this point in the revised paper (in Remark 1 of Section 4.1). A safe conclusion to draw is from the empirical perspective: “The experimental results in Appendix B.4 show that training ROBOT can significantly decrease the training robust loss while directly optimizing the robust loss fails to do so. This indicates that the reparametrization of ROBOT may have better convergence property on minimizing the robust loss.” An in-depth investigation of the mechanism needs more careful experimental designs and theoretical analysis, which are our future work.

---

> > > > ### Comment · Reviewer_FXEG · 2022-11-22
> > > > **Thank you for the update**
> > > >
> > > > I'm satisfied with the authors' answers.

---

### Official Review · Reviewer_Ukut · 2022-10-29

**Confidence:** 3
**Correctness:** 3
**Technical Novelty And Significance:** 3
**Empirical Novelty And Significance:** 3
**Recommendation:** 6

**Clarity, Quality, Novelty And Reproducibility:**

Clarity: Mostly clear.
Quality: Good.
Reproducibility: Code and proofs are there, but not fully sure of reproducibility.

**Strength And Weaknesses:**

Strengths:
The paper addresses an important problem of learning with noisy labels. Though I am not super familiar with the literature on noise transition matrix estimation, the characterization of existing works under a simple framework (MGEO) seems nice. Then, they highlight the problems with existing works -- anchor point assumptions and in-consistency when noise posterior estimation is imperfect. In addition the paper also gives a new solution to this problem and it is shown that the proposed solution has nice properties like identifiability, consistence and generalization. Empirical evaluation on multiple datasets shows significant improvement over the other baselines.

Weaknesses/Questions:
Could you please provide some background/preliminaries on the methods that are being put under MGEO? For the readers who are not super familiar with this line of work. I could see a formal statement on the in-consistency of MGEO methods, it is discussed using an example in section 4. Is it possible to have an exact statement on the in-consistency -- even restating previous results could be helpful. In Theorem 2 why are the results in terms of the covering number of $\mathcal{F}$? Can't the standard generalization error bounds be instantiated here? What is $\epsilon$ in Theorem 2? Is there a bound on the estimation error of $T$  and do you need any assumptions on the noise level (original $T$) for the method to work? I believe if the noise level is too high then the estimation might be hard, is it true?


**Summary Of The Paper:**

This paper studies the problem of estimating instance independent noise transition matrix ($T$), which is utilized in learning with noisy labels. They unify existing methods for $T$ estimation, under a framework called Minimum Geometric Envelope Operator and show that these methods are prone to failing ( they are in-consistent estimators) when the noise posterior estimation is imperfect. To address this issue, they propose a novel T-estimation framework (ROBOT) based bilevel optimization and show that their proposal has nice theoretical properties like identifiably, consistency and finite-sample generalization guarantees. They also argue that unlike the MEGO methods their method doesn't require assumptions on the existence of anchor points. Empirical evaluation on multiple benchmarks shows the superiority of the proposed method.

**Summary Of The Review:**

Overall a good paper. It provides an abstraction and highlights issues with existing methods for estimation of noise transition matrix and also proposes a solution to fix those issues.

---

> ### Author Response · Authors · 2022-11-15
> **Author Response**
>
> > "*Could you please provide some background/preliminaries on the methods that are being put under MGEO?*"
>
> Re: We have added a section in Appendix A.1 to describe the existing methods, e.g., anchor based and anchor free (minimum volume) methods. Adding this part indeed makes the paper more self-contained. Thanks for the great suggestion!
>
> > "*For the readers who are not super familiar with this line of work. I could see a formal (informal?) statement on the in-consistency of MGEO methods, it is discussed using an example in section 4. Is it possible to have an exact statement on the in-consistency -- even restating previous results could be helpful.*"
>
> Re: Thanks for the question!  We add a formal result on the in-consistency of MGEO. We put it in Corollary 1 in Section 3 of the revised paper. Corollary 1 shows that when the posterior is not perfect, the estimated $T$ of MGEO won’t converge to the oracle $T^*$ when the sample size goes to infinity. Corollary 1 is an exact statement of the in-consistency of MGEO. The proof of Corollary 1 is a direct application of  Proposition 1’s proof. Adding Corollary 1 indeed makes our paper clearer. Thanks again!
>
> > "*In Theorem 2 why are the results in terms of the covering number of F?  Can't the standard generalization error bounds be instantiated here?*"
>
> Re: This is a good question! The standard generalization technique is suitable when the function class is discrete (finite). It is difficult to apply the standard generalization technique on a function class with infinite elements because the uniform convergence needs to hold for each element of the function class. In our case, the function class $F$ induced by $T$ is continuous and it contains infinite elements. So we need to use a covering technique, which splits the space into a finite number of pieces. The width of each piece is bounded by $\epsilon$. This ensures that for any function $f$ in the original continuous space, we can find always an element $f'$ in the discrete space which is $\epsilon$-close to $f$. So the uniform convergence analysis can be transformed to the discrete space by paying an additional error term $\epsilon$. Notably, we can also apply techniques like VC-dimension or Radamarcher Complexity to replace the covering technique. They can lead to similar results.
>
> > "*What is ϵ in Theorem 2?*"
>
> Re: For convenience, we use $\epsilon$ for both the sub-optimality of the optimization and the covering interval in the analysis. The $\epsilon$ is any small positive and fixed value (we add this description in Theorem 2). A small $\epsilon$ also indicates that we solve the optimization well and decrease the training loss to the $\epsilon$-optimality. We also construct a  $\epsilon$-covering of the continuous function space. Notably, we can construct the covering of the function space at any small interval. We choose the interval as $\epsilon$ here to simplify the results. If we use $\epsilon_1$ for the sub-optimality of the optimization and  $\epsilon_2$ for the covering interval in the analysis, the $2\epsilon$ in the equation 8 will be replaced by $\epsilon_1 + \epsilon_2$.
>
> > "*Is there a bound on the estimation error of T and do you need any assumptions on the noise level (original T) for the method to work? I believe if the noise level is too high then the estimation might be hard, is it true?*"
>
> Re: **The bound on the estimation error of T**. Thanks for the question! We add an additional theoretical result in Appendix A.11 in the revised version. We directly show that $T$ converges to $T^*$ with high probability under suitable conditions. Thanks for the suggestion! This makes our paper more complete.
>
> **The requirement for the noise ratio**. Yes, you are absolutely right! We indeed need an assumption on the noise level. This is implicitly required by the robust loss’s property in equation (6). For example, the reverse cross entropy and mean absolute loss both need the noise ratio $\eta$ to be smaller than $1 - 1/K$ when the noise is symmetric. Here the $K$ is the number of classes.  We add more description in Appendix A.2 in the revised version, including conditions on symmetric and class-dependent noises. Thank you for pointing this out!

---

> > ### Author Response · Authors · 2022-11-22
> > **Further Discussion**
> >
> > Dear Reviewer Ukut:
> >
> > Thanks for your insightful reviews and constructive comments. Your suggestions indeed help us to improve our paper. Could you kindly take a look at the responses to see whether they can address your concerns and improve the paper? Here is a summary of our response and revision:
> >
> > *  We add more descriptions of the existing methods (anchor based and anchor free methods), robust losses, and forward correction losses (including their working conditions) in Appendix A.1-A.3 to make our paper more self-contained.
> > *  We add a formal result on the in-consistency of MGEO in Corollary 1 (Section 3 of the revised paper).
> > * We add an additional theoretical result in Appendix A.11 on directly bounding the estimation error of T.
> > * We discuss the value $\epsilon$ in Theorem 2 and why we use the covering number to derive the uniform convergence in the response.
> >
> > We would be grateful if you could kindly take a look at both the revision and our response. Please let us know if there are any further questions or suggestions that we could clarify or improve.
> >
> > Sincerely
> >
> > The authors

---

### Official Review · Reviewer_nwAC · 2022-10-30

**Confidence:** 2
**Correctness:** 4
**Technical Novelty And Significance:** 3
**Empirical Novelty And Significance:** 3
**Recommendation:** 8

**Clarity, Quality, Novelty And Reproducibility:**

While the grammar and expressions are clear, the exposition of the method is a bit convoluted. The paper could be significantly strengthened by an overview of the proposed approach and an explanation of how each term is implemented. As far as I can tell, the proposed approach is novel.

**Strength And Weaknesses:**

Strengths

1. Handling label noise in deep learning is a significant and practically important problem as it is well-known that obtaining good posteriors from deep learning models is a challenge.  The propsed method is therefore a welcome contribution in this direction.

2. The unified view of anchor-based and anchor-free, MEGO, methods provides some interesting insights into the existing approaches for handling noisy labels and is a good way of summarizing these approaches.

3. The proposed method for overcoming the limitations of MEGO has good theoretical guarantees which are described in great detail in the paper.

4. The empirical results show that the method consistently outperforms existing approaches across a range of datasets.

Weaknesses

1. While the proposed method is quite simple, the paper itself is a bit difficult to read and follow. For example, there is no summary of the approach and some implementation details aren’t mentioned. How is $\mathcal{L}_{rob}$ computed?

2. The evaluation is on rather small datasets that don’t really have much practical significance in the era of large models. How does the method scale in terms of dataset size and how computationally expensive is the method?

**Summary Of The Paper:**

The contributions of this paper are two-fold: firstly, a unifying view of the noise transition matrix is presented which combines both anchor-based and anchor-free methods into a single framework called MEGO which is based on the minimum-volume convex hull. The authors then propose an improved bi-level optimization method called ROBOT which improves the performance of the method in the presence of imperfect posterior estimation. The results show that the method outperforms the baselines by some margin.



**Summary Of The Review:**

Overall, this paper makes a good contribution both in terms of providing a unified view of existing noise transition matrix estimation approaches and proposing a new effective method for dealing with noisy posterior estimation. The only major weakness is there is no evaluation of the significance of the approach on real-world large scale datasets.

---

> ### Author Response · Authors · 2022-11-15
> **Author Response**
>
> > "*While the proposed method is quite simple, the paper itself is a bit difficult to read and follow. For example, there is no summary of the approach and some implementation details aren’t mentioned. How is Lrob computed?*"
>
> Re: Thanks a lot for the advice! We add a summary of our method and a detailed description of the implementations in Appendix A.8 and A.10 of the revised paper. For the computation of $L_{rob}$, it is shown in Appendix A.10. We also add more descriptions of the existing methods (anchor based and anchor free methods), robust losses, and forward correction losses in Appendix A.1-A.3 to make the paper more self-contained and easier to follow.
>
> > "*The evaluation is on rather small datasets that don’t really have much practical significance in the era of large models. How does the method scale in terms of dataset size and how computationally expensive is the method?*"
>
> Re: Thanks for the question! Table 5 shows the results of our method on Clothing1M dataset, which contains 1 million clothing images in 14 classes. To the best of our knowledge, this is one of the largest benchmark datasets for noisy label learning problems in the existing literature. We use ResNet50 for Clothing1M. Our method works efficiently on such a large-scale dataset. Actually, we do not need to exactly solve the inner loop of equation 7. We use a one-step approximation [1, 2] for the inner problem during implementation (summarized in Appendix A.8), which alternatively updates the $T$ (outer loop) and network parameters (inner loop). With this approximation technique, the overall computational cost is as low as 1.6 times the cost for regular training on Clothing1M. Note that these techniques have been widely used in previous works and demonstrated superior performance [1, 2].
>
> [1] Jun shu, et. al., Meta-Weight-Net: Learning an Explicit Mapping For Sample Weighting
>
> [2] Amirreza Shaban, et. al., Truncated Back-propagation for Bilevel Optimization, AISTATS 2019
>
> > "* The paper could be significantly strengthened by an overview of the proposed approach and an explanation of how each term is implemented.*"
>
> Re: Thanks for the suggestion! We have included the summary of our algorithm and its implementation details in Appendix A.8 and A.10 in the revised version. Adding these parts indeed improves our paper's clarity and completeness!

---

### Official Review · Reviewer_5YUk · 2022-10-31

**Confidence:** 2
**Correctness:** 3
**Technical Novelty And Significance:** 3
**Empirical Novelty And Significance:** 2
**Recommendation:** 6

**Clarity, Quality, Novelty And Reproducibility:**

* The clarity of the paper could be improved
* The method is certainly novel and easy to reproduce

**Strength And Weaknesses:**

Strength

* The paper is backed by enough theoretical evidence.
* Results are strong.

Weakness

* In the last sentence in page-5, text(T) was used with out defining the term.
* If I understood the eq 7 correctly, one needs to optimise the objective and the constraint in an iterative manner.  The neural network parameters are optimised using the noisy training set and the current estimation of the transition matrix. The transition matrix is then updated using the validation set. It is not clear why one would need to divide the set into training and validation and why can not we use the same noisy annotation to estimate both iteratively?
* The theorem 2 is valid if eq-7 is well solved. I could not find any convergence guarantees of eq-7.



**Summary Of The Paper:**

To construct the noise probability matrix, existing methods assume that there exists anchor points which belong to a certain class probability one or assume a perfect posterior estimator for the noisy examples. A new transition matrix estimation is proposed that utilizes an alternate optimisation routine of the neural network and the transition matrix parameters.

**Summary Of The Review:**

The paper shows enough theoretical evidence of the method. However, some parts will not be easy to be accessible by non-expert readers.

---

> ### Author Response · Authors · 2022-11-15
> **Author Response**
>
> > "*In the last sentence in page-5, text(T) was used without defining the term.*"
>
> Re: Thanks for pointing this out! Actually, it is a typo. It should be $\textup{conv}(T)$, which is the convex hull formed by the columns of $T$. We have fixed this in the revised paper.
>
> > "*It is not clear why one would need to divide the set into training and validation and why can not we use the same noisy annotation to estimate both iteratively?*"
>
> Re: This is a good question! Actually, the following two methods perform similarly: 1) training the bilevel framework on the split dataset (splitting into training and validation datasets); 2) training the bilevel framework on the same dataset for the inner and outer loop. We have tried them both and found their performances to be close. The experiments and results are included in  Appendix B.5 of the revised paper.
>
> The reason why we present ROBOT in the split dataset manner is as follows:
>
> - It is a common practice in bilevel methods [1];
> - It is convenient for theoretical analysis. The reason is as follows: $\hat \theta (T)$ obtained on the training dataset is independent of the validation dataset $\mathcal{D}^n_v$. In this case, when $T$ is fixed, we can simply use standard Hoeffding inequality to bound the population loss $L_{rob}(\hat \theta (T), D)$ by the finite sample loss $L_{rob}(\hat \theta (T), {D}^n_v)$.
>
> On the practical side, we can either choose to split the dataset or not.
>
> [1] Jun shu, et. al., Meta-Weight-Net: Learning an Explicit Mapping For Sample Weighting
>
> > "*The theorem 2 is valid if eq-7 is well solved. I could not find any convergence guarantees of eq-7.*"
>
> Re: We have added the optimization convergence of eq-7  in Appendix A.9 of the revised paper. The convergence is first established in [1] and we restate it for ROBOT. Adding this indeed makes our paper more complete. Thanks for the great suggestion!
>
> [1] Amirreza Shaban, Ching-An Cheng, Nathan Hatch, Byron Boots, Truncated Back-propagation for Bilevel Optimization, AISTATS 2019
>
> > "*However, some parts will not be easy to be accessible by non-expert readers.*"
>
> Re: Thank you for the question! We have added more descriptions of the backgrounds and existing methods (anchor based and anchor free methods) in Appendix A.1. We also discuss the robust losses and the forward correction method in Appendix A.2 and A.3. We hope that adding these parts can make the paper more self-contained and easier to read.

---

> > ### Author Response · Authors · 2022-11-22
> > **Further Comment and Discussion**
> >
> > Dear Reviewer 5YUk,
> >
> > Thank you for your valuable time to review our work and for your constructive feedback! We posted our response to your comments several days ago, and we wonder if you could kindly share some of your thoughts so we can keep the discussion rolling to address your concern if there are any.
> >
> > In the previous response:
> > * We add convergence guarantee of eq-7  Appendix A.9.
> > * We have added more descriptions of the backgrounds and existing methods in Appendix A.1-A.3 for better readability and completeness.
> > * We discuss that splitting the dataset into training and validation datasets is convenient for theoretical analysis. In practice, we can either split the dataset or use the same dataset for both inner and outer loops in eq-7, since they result in similar performance as shown in Appendix B.5.
> >
> > We humbly expect you could check our responses with our updated version. More discussions are always welcome.
> >
> > Sincerely
> >
> > The authors

---

### Official Review · Reviewer_NebR · 2022-11-02

**Confidence:** 1
**Correctness:** 4
**Technical Novelty And Significance:** 3
**Empirical Novelty And Significance:** Not applicable
**Recommendation:** 8

**Clarity, Quality, Novelty And Reproducibility:**

In spite of my efforts I am unable to provide an acceptable review of this paper.


**Strength And Weaknesses:**

In spite of my efforts I am unable to provide an acceptable review of this paper.




**Summary Of The Paper:**

In spite of my efforts I am unable to provide an acceptable review of this paper.

**Summary Of The Review:**

In spite of my efforts I am unable to provide an acceptable review of this paper.

---

> ### Author Response · Authors · 2022-11-15
> **Author Response**
>
> Thanks for your effort anyway! The message for us is that we need to make our paper easier to read. We add more descriptions of the existing methods (anchor based and anchor free methods), robust losses, and forward correction losses (including their working conditions) in Appendix A.1-A.3; we also add a summary of our method and a detailed description of the implementations in Appendix A.8 and A.10. If there is any question, we are very glad for further discussions.

---

### Author Response · Authors · 2022-11-15
**General Author Response**

We thank all the reviewers for their great effort and insightful reviews - they will certainly help improve the clarity, presentation, and completeness of our paper! We are also glad that all reviewers are in agreement about the novelty, theoretical insights, and good empirical performance of the paper.

We have uploaded a revised version of our paper following the suggestions/comments from all the reviewers. Some major changes are:

- We add more descriptions of the existing methods (anchor based and anchor free methods), robust losses, and forward correction losses (including their working conditions) in Appendix A.1-A.3; we also add a summary of our method and a detailed description of the implementations in Appendix A.8 and A.10. Thanks for 5YUk*,* nwAC, Ukut, and FXEG's advice!
- We add additional theoretical results including 1) a formal statement of the in-consistency of MGEO in Corollary 1 in Section 3; 2) the convergence guarantees (from the optimization perspective) of Equation-7 in Appendix A.9; 3) a direct bound on the $T$ estimation error in Appendix A.11. Thanks for Ukut and 5YUk's advice!
- We add additional experimental results in Appendix B.4 to show that adopting ROBOT can significantly reduce the difficulty of optimizing the robust loss. We show that ROBOT can effectively decrease the robust loss on the training dataset while direct optimization can not. We further provide more discussion on the mechanism behind it. Thanks for FXEG's advice!
- We add additional experiments in Appendix B.5 to compare the following two methods: 1) training ROBOT on the split dataset (splitting into training and validation datasets, the training set for the inner loop and the testing set for the outer loop); 2) training ROBOT on the same dataset for the inner and outer loop. These two schemes lead to similar performance. Thanks for 5YUk's advice!

We thank the reviewers again and look forward to hearing their thoughts after reading the response!

---

### Decision · Program_Chairs · 2023-01-20

**Decision:**

Accept: notable-top-25%

**Justification For Why Not Higher Score:**

Clarity needs to be improved

**Justification For Why Not Lower Score:**

New interesting algorithm for an important problem in ML.

**Metareview: Summary, Strengths And Weaknesses:**

The paper investigates learning statistically consistent classifiers under label noise. Here, the authors present a new transition matrix estimation that utilizes an optimisation routine of the neural network and the transition matrix parameters. Experiments show that this approach yields superior performance.

The paper offers a good motivation and tackles an important problem. The algorithm is well motivated and derived by a recent amount of theory. There were initially some concerns about the scalability in terms of the experiments as well as the readability of the paper. Both concerns have been addressed properly in the rebuttal.

**Note From Pc:**

if the above contains the word "oral" or "spotlight" please see: "oral" presentation means -> notable-top-5% and "spotlight" means -> notable-top-25%. As stated in our emails, we are disassociating presentation type from AC recommendations

**Summary Of Ac-Reviewer Meeting:**

NA